# Global-Recent Semantic Reasoning on Dynamic Text-Attributed Graphs with Large Language Models

**Yunan Wang, Jianxin Li, Ziwei Zhang**[*]

School of Computer Science and Engineering, Beihang University

`zwzhang@buaa.edu.cn`

## Abstract

Dynamic Text-Attribute Graphs (DyTAGs), characterized by time-evolving graph interactions and associated text attributes, are prevalent in real-world applications. Existing methods, such as Graph Neural Networks (GNNs) and Large Language Models (LLMs), mostly focus on static TAGs. Extending these existing methods to DyTAGs is challenging as they largely neglect the *recent-global temporal semantics*: the recent semantic dependencies among interaction texts and the global semantic evolution of nodes over time. Furthermore, applying LLMs to the abundant and evolving text in DyTAGs faces efficiency issues. To tackle these challenges, we propose Dynamic Global-Recent Adaptive Semantic Processing (DyGRASP), a novel method that leverages LLMs and temporal GNNs to efficiently and effectively reason on DyTAGs. Specifically, we first design a node-centric implicit reasoning method together with a sliding window mechanism to efficiently capture recent temporal semantics. In addition, to capture global semantic dynamics of nodes, we leverage explicit reasoning with tailored prompts and an RNN-like chain structure to infer long-term semantics. Lastly, we intricately integrate the recent and global temporal semantics as well as the dynamic graph structural information using updating and merging layers. Extensive experiments on DyTAG benchmarks demonstrate DyGRASP's superiority, achieving up to 34% improvement in Hit@10 for destination node retrieval task. Besides, DyGRASP exhibits strong generalization across different temporal GNNs and LLMs. Code available at `https://github.com/Nala-YN/DyGRASP`.

## 1 Introduction

Dynamic Text-Attribute Graphs (DyTAGs) are widely present in real-world scenarios like E-commerce, knowledge graphs, and social networks (Khrabrov & Cybenko, 2010; Deng et al., 2019; Song et al., 2019; Zhang et al., 2022; Tang et al., 2023; Luo et al., 2023; Huang et al., 2022). Unlike commonly studied TAGs (Sen et al., 2008; Giles et al., 1998; Mernyei & Cangea, 2020; Hu et al., 2020; He et al., 2023; Yan et al., 2023), where nodes only contain static text attributes, DyTAGs contain evolving information over time and involve interactions accompanied by timestamp and dynamic text attributes (Zhang et al., 2024). For example, consider an E-comment graph illustrated in Figure 1 where nodes represent users or merchants and interactions represent user reviews to merchants. Clearly, the spatio-temporal patterns of DyTAGs contain more abundant information than static TAGs. How to fully mine the rich value underlying DyTAGs is essential for both academia and industry.

For static TAGs, most primitive approaches resort to Graph Neural Networks (GNNs) (Wu et al., 2019; Li et al., 2021; Veličković et al., 2018; Hamilton et al., 2017; Kipf & Welling, 2017) for their end-to-end learning capabilities. While excelling at capturing graph structural information, these methods usually only adopt shallow text encodings, such as Bag-of-Words or word embeddings as features, thus lacking strong semantic knowledge to understand the textual attributes comprehensively. Recently, Large Language Models (LLMs) (Abdin et al., 2024; Almazrouei et al., 2023; Touvron et al.,

---

[*]Corresponding Author.

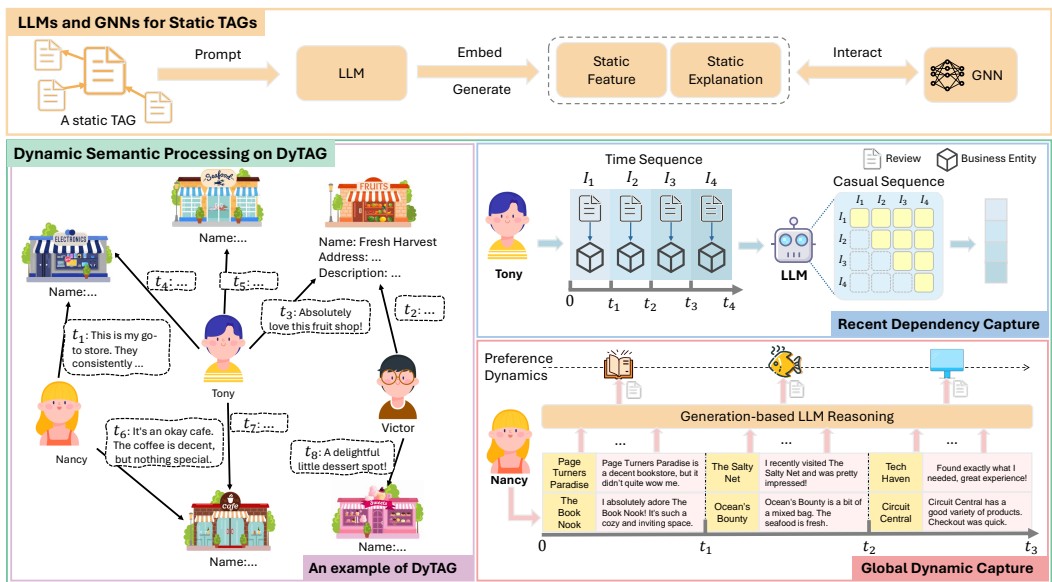

Figure 1: An illustrative comparison between handling static TAGs and DyTAGs. Static methods that utilize LLMs usually generate static embeddings or explanations for text attributes. In comparison, DyTAGs contain rich spatio-temporal information, such as recent semantic dependency or global semantic dynamics of nodes, which the existing methods for TAGs do not account for.

2023; Guo et al., 2025; Achiam et al., 2023), pretrained on vast text corpora, have demonstrated strong capabilities in text understanding and generation. Thus, researchers have explored various methods to combine GNNs and LLMs for TAGs (Zhu et al., 2025; Wang et al., 2024b; Khoshraftar et al., 2025; Beiranvand & Vahidipour, 2025; Chen et al., 2024), aiming to integrate textual information into graph structures. Despite the remarkable progress in combining LLMs and GNNs for TAGs, these methods cannot be directly transferred to DyTAGs due to the following challenges:

Firstly, existing GNNs and LLMs neglect the global-recent semantic dynamics in DyTAGs. In real life, the temporal changes of many phenomena conform to a complicated mixture of recent and global patterns. For example, ice cream sales might be related to recent promotional activities while also being influenced by long-term dietary structure changes. Specifically, we investigate two complementary temporal semantic features with different temporal granularities within DyTAG:

- **Recent semantic dependency of temporal interactions:** The text attribute of an interaction in DyTAG exhibits recent semantic dependency. For instance, the word "notebook" might mean "notepad" following a "visit bookstore" interaction, but the same word is more likely to mean "computer" after a "visit electronics store" interaction.

- **Global semantic dynamics of nodes:** Unlike static node semantics, node features in DyTAG also continuously undergo global changes, which are reflected by emerging interactions. For example, a user in an E-commerce network may shift her interests from literature to technology, as reflected in visiting different kinds of shops.

Current GNNs and LLMs methods for TAGs largely ignore the above issues as they only employ static methods to model the semantic and structural information. .

Secondly, DyTAGs pose more severe efficiency challenges for LLMs. For static TAGs, most text attributes are associated with nodes. However, for DyTAGs, there exist edge-level text (such as the example in Figure 1) and the edge attributes can also evolve with time. Considering that the number of edges is usually orders of magnitude larger than the number of nodes and there can exist at least hundreds or thousands of time stamps, developing effective models for DyTAGs demands highly efficient LLM reasoning designs to reduce computational costs.

To tackle these challenges, in this paper, we propose DyGRASP(Dynamic Global-Recent Adaptive Semantic Processing), which leverages both the implicit and the explicit reasoning capabilities of LLM to capture recent-global temporal semantics in DyTAGs. Specifically, DyGRASP features three novel designs: (i) Implicit reasoning for recent temporal semantics: to capture recent semantic dependency

while retaining a low computational complexity, we leverage the unidirectional consistency between temporal sequences and causal sequences of LLMs to achieve node-centric implicit reasoning. In addition, we propose a sliding window mechanism to organize a node's historical interactions chronologically in batches for the LLM. Our proposed method models recent semantic features for multiple interactions simultaneously while ensuring no future information leakage, which significantly reduces the number of consumed tokens so that our method can be scaled to real-world DyTAGs; (ii) Explicit reasoning for global temporal semantics: to learn node representations from extensive historical interactions while reducing consumed number of tokens, we utilize the generative reasoning ability of LLM and segment a node's historical interactions into a constant number of partitions based on timestamps. Then, we carefully design prompts to instruct the LLM to summarize a node's feature description for each period. Finally, we employ an RNN-like reasoning chain structure to pass long-distance semantics and dynamically update node features; (iii) Integrating Semantics and Graph Structure: lastly, we design three tailored layers to update recent semantics, global semantics, and graph structural features and then integrate them, together with a temporal GNN to learn comprehensive node representations.

To verify DyGRASP's effectiveness, we conduct extensive experiments on DyTAGs benchmark. Our proposed method achieves up to a 34% improvement in Hit@10 for destination node retrieval task compared to state-of-the-art methods. Besides, DyGRASP demonstrates strong generalization across different LLMs and Temporal GNNs. We also perform detailed analyses for different components and hyperparameters. Our contributions are summarized as follows:

- We study the recent-global spatio-temporal patterns in DyTAGs, which are overlooked in existing GNNs and LLMs for static TAGs.
- We propose to leverage the implicit and explicit reasoning capabilities of LLMs to capture the recent-global semantics on DyTAGs. Motivated by this goal, we propose DyGRASP, a tailored model fusing the advantages of LLMs and temporal GNNs.
- We theoretically prove that our proposed method has optimized reasoning efficiency compared to straightforward methods, thereby reducing the cost of practical applications.
- We verify the effectiveness of DyGRASP through extensive experiments on DyTAG benchmarks, outperforming state-of-the-art baselines up to 34%.

## 2 RELATED WORK

**LLM for TAGs.** While GNNs excel at encoding graph structure features, preliminary methods exhibit deficiencies in understanding the textual content within TAGs. Consequently, researchers leverage the powerful capability of LLMs on text understanding to enhance the performance of GNNs on TAGs (Chien et al.; Yang et al., 2021; Xue et al., 2023; Wu et al., 2024; Wei et al., 2024; Guo et al., 2024; Liu et al., 2024). Among them, ENGINE (Zhu et al., 2024) employs LLMs to enhance the quality of node representations by proposing "G-Ladder" structure. TAPE (He et al., 2024) utilizes "explanations" generated by an LLM for the textual attributes as features. SimTeG (Duan et al., 2023) boosts textual graph learning by first fine-tuning a language model to generate node embeddings, which are then used as features for a separate GNN. GraphGPT (Tang et al., 2024) employs graph instruction tuning to integrate LLMs with graph structural knowledge. However, the aforementioned methods primarily target TAGs containing only static semantic features. The dynamic characteristics of DyTAGs render these approaches incapable of modeling the temporal semantic relationships therein. In contrast, we investigate the temporal semantics inherent in DyTAGs by leveraging the reasoning capabilities of LLMs.

**Temporal Graph Neural Networks**. To extend the capabilities of GNNs to dynamic graphs, recent studies have proposed various temporal GNNs (Rossi et al., 2020; Poursafaei et al., 2022; Wang et al., 2021b; Luo & Li, 2022; Ma et al., 2020; Cong et al., 2023; Xu et al., 2025; Liu et al., 2025a;b) to model dynamic graph features. Among these, TGAT (Xu et al., 2020) introduces functional time encoding based on Bochner's theorem. CAWN (Wang et al., 2021c) leverages temporal random walks to inductively learn representations of temporal network dynamics. DyRep (Trivedi et al., 2019) learns dynamic graph representations by modeling a latent mediation process between topological evolution and node activities. DyGFormer (Yu et al., 2023) learns solely from the historical first-hop interaction sequences of nodes to simplify the dynamic graph learning task. CTAN (Gravina et al., 2024) captures long-range dependencies in continuous-time dynamic graphs via non-dissipative ODEs. TMetaNet

(Li et al., 2025) leverages Dowker Zigzag Persistence to guide meta-learning parameter updates for dynamic link prediction. Mspace (Rahman & Coon, 2025) models spatiotemporal dynamics using a Markov-based approach for node feature forecasting. However, the aforementioned temporal GNNs are unable to comprehend the textual semantic information within DyTAG, leading to suboptimal performance. In comparison, our method introduces LLMs reasoning over DyTAG to capture the temporal semantic features.

**LLM for DyTAGs.** Following the pioneering benchmark for DyTAGs named DGTB (Zhang et al., 2024), there has recently been few research on LLMs for DyTAGs, which are largely concurrent to our work. Among them, LKD4DyTAG (Roy et al., 2025) distills knowledge from LLM into a temporal GNN. CROSS (Zhang et al., 2025) employs LLMs to dynamically extract text semantics and a co-encoder to synergistically unify these semantics with evolving graph structures. GAD (Lei et al., 2025) utilizes a multi-agent system with collaborative LLMs to directly perform prediction on DyTAGs without dataset-specific training. However, existing studies has not addressed the reasoning efficiency challenges when applying LLMs to DyTAG and are unable to identify the recent-global patterns inherent in DyTAG, thus failing to comprehensively capture its multi-granularity temporal semantic features.

## 3 NOTATIONS

**Definition 1 (DyTAG)** *A DyTAG is represented as $\mathcal{G} = \{\mathcal{V}, \mathcal{E}, \mathcal{T}, \mathcal{D}, \mathcal{R}\}$, where $\mathcal{V}$ is the set of nodes, $\mathcal{E} \subseteq \mathcal{V} \times \mathcal{V}$ is the set of edges, $\mathcal{T}$ is the set of observed timestamps, $\mathcal{D}$ is the set of node textual attributes, and $\mathcal{R}$ is the set of edge textual attributes. The textual attribute of a node $u$ is denoted by $D_u \in \mathcal{D}$. An edge, also known as an interaction, occurring between nodes $u$ and $v$ at timestamp $t$ is represented as $I = (u, \mathbf{r}, v, t)$, where $\mathbf{r} \in \mathcal{R}$ represents the textual attribute.*

We use interaction to refer to an edge in DyTAGs to emphasize its dynamic nature, e.g., interaction text denotes the complete description of the source node's behavior towards the target node, rather than merely the edge's text attribute itself. We use reasoning to represent the LLM's process of uncovering underlying information from a series of interactions. We aim to develop models that can handle various tasks associated with DyTAGs, such as node classification, link prediction, and destination node retrieval.

## 4 METHOD

In this section, we introduce our method. We first introduce implicit reasoning for recent temporal semantics in Section 4.1, then introduce explicit reasoning for global temporal semantics in Section 4.2, and finally integrate the temporal semantics with the dynamic graph structures in Section 4.3.

### 4.1 IMPLICIT REASONING FOR RECENT TEMPORAL SEMANTICS

Unlike the static attributes in normal TAGs, the textual attributes in DyTAG contain temporal semantic relationships with historical interactions. Therefore, we leverage the implicit reasoning ability of LLMs by harvesting the hidden embedding of texts within LLMs.

To obtain textual attribute embeddings that incorporate temporal semantics, an intuitive approach is to take an edge-centric view and organize the textual attributes of an interaction and historical interactions chronologically, and use an LLM to extract the hidden embeddings for the interaction. However, denoting $d$ as the average degree of the graph, the complexity (i.e., the number of consumed input tokens) of the LLM reaches $O(|\mathcal{E}| \times d)$. For a real-world DyTAG that has millions of edges and an average degree of hundreds or thousands, this straightforward method suffers from an excessively high computational complexity. To address this potential issue while preserving temporal semantic relationships, we propose two key designs: (i) node-centric implicit reasoning, which leverages the unidirectional consistency between the causal order inherent in LLMs and the chronological order of the dynamic graph; (ii) sliding window mechanism, which segments interaction sequences into overlapping batches to balance computational consumption and semantic feature modeling capability. Using these two designs, our proposed method can theoretically reduce the computational complexity to $O(|\mathcal{E}|)$ (Proof is shown in Appendix C). Next, we elaborate on these two designs.

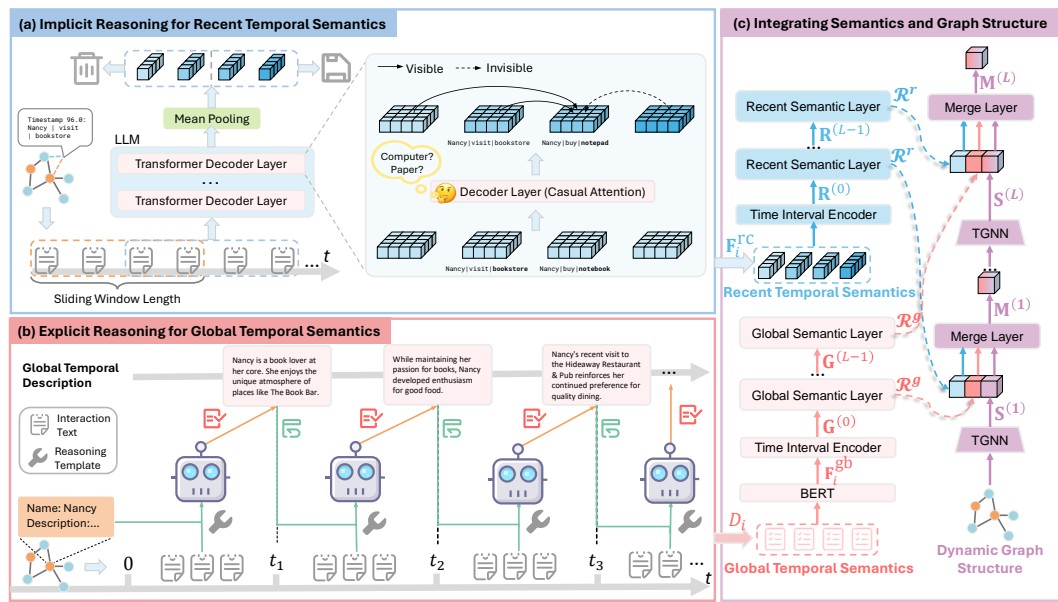

Figure 2: An overview of DyGRASP. DyGRASP first utilizes the implicit and explicit reasoning capabilities of the LLM to respectively extract recent and global temporal semantic features from a DyTAG, as shown in (a) and (b), respectively. Then it integrates these temporal semantic features with the dynamic graph structure features by temporal GNN, as shown in (c).

**Node-centric implicit reasoning.** As illustrated in Figure 2(a), for a node $v$ in $\mathcal{G}$, we first extract all its interactions with its neighbors, denoted as $\mathcal{N}_v$. Then we sort them by the timestamp and arrange them into a sequence, which ensures interactions earlier in time positioned earlier in the sequence. Lastly, we input $\mathcal{N}_v$ into an LLM at once to extract hidden features of $\mathcal{N}_v$. As LLMs are designed for causal modeling of text sequences and share the unidirectional nature with the time sequence of $\mathcal{N}_v$, this consistency prevents an interaction from observing future interaction text, thereby avoiding leakage of future information. Benefiting from this, in a single forward pass, all interactions in $\mathcal{N}_v$ can simultaneously acquire recent semantic features from the historical interactions thus enabling efficient reasoning.

**Sliding window mechanism.** This module aims to solve three challenges: (i) the context window length limitation of LLMs, (ii) the heavy computational resource consumption, (iii) the tendency for LLM performance to degrade with increased input length (Wang et al., 2024a; Levy et al., 2024). Specifically, we partition $\mathcal{N}_v$ into multiple overlapping batches:

$$\mathcal{B}_i = \{I_k | \frac{c}{2} * i + 1 \le k \le \frac{c}{2} * i + c, I_k \in \mathcal{N}_v\}, \tag{1}$$

where $\mathcal{B}_i$ is the $i$-th batch and $c$ is the window length. In each batch, the first $\frac{c}{2}$ interactions serve to provide temporal context for the subsequent $\frac{c}{2}$ interactions. Subsequently, for each batch, we organize the interactions chronologically into an input sequence using a dataset-specific prompt template (see an example in Figure 6) and feed it into the LLM. Finally, we apply mean pooling to the output hidden layer features for each interaction. The obtained temporal semantic feature for an interaction $i \in I$ is denoted as $\mathbf{F}_i^{\text{rc}} \in \mathbb{R}^{d_{\text{LLM}}}$, where $d_{\text{LLM}}$ denotes the dimensionality of the hidden features.

## 4.2 EXPLICIT REASONING FOR GLOBAL TEMPORAL SEMANTICS

Complementary to recent temporal features, the occurrence of a real-world phenomenon is often associated with long-range global temporal features. Therefore, this module aims to uncover these global temporal semantic features with the explicit reasoning ability of LLMs, i.e., we directly let LLMs generate textual information after comprehending DyTAGs.

Specifically, as shown in Figure 2(b), we first evenly partition the interaction sequence $\mathcal{N}_v$ into $s$ segments by timestamps. Each segment is denoted by $S_i(1 \le i \le s)$. The partition timestamps are denoted by $\mathbf{T}^{\text{gb}} = \{\hat{t}_0, \hat{t}_1, \hat{t}_2, ..., \hat{t}_s\}$, where $\hat{t}_0 = -1$ denotes the special starting time. $S_i$ consists

of the chronologically ordered sequence of all interactions occurring between $\hat{t}_{i-1}$ (exclusive) and $\hat{t}_i$ (inclusive). Our motivation is twofold: (i) similar to implicit reasoning, $\mathcal{N}_v$ cannot be fed into LLM at once; (ii) performing only a single inference fails to capture the temporal evolution of global semantic features.

Next, we employ an RNN-like reasoning chain structure to empower the LLM with memory capabilities. Specifically, let $D_i(1 \leq i \leq s)$ represent the textual description of the node's dynamic features corresponding to the period of $S_i$, and $D_0$ be the node's original textual attribute. The reasoning process can be described as:

$$D_i = \text{LLM}\left(D_{i-1}, S_i\right), \tag{2}$$

where we use dataset-specific template (see Figure 7 for an example) to prompt LLM to uncover the potential global semantics. The reasoning chain structure is designed for the propagation of long-range dynamic features and providing essential background description of the node for the LLM to infer the node's current features.

Crucially, this reasoning process is designed to be cumulative and iterative. The input $D_{i-1}$ serves as a cumulative summary of the node's entire interaction history up to the start of the current segment $S_i$, rather than a local feature of the previous segment alone. Consequently, the LLM is tasked with updating this historical summary by integrating the new interactions contained in $S_i$. For instance, $D_1$ summarizes $S_1$, while $D_2$ builds upon $D_1$ to incorporate $S_2$, thereby summarizing the combined history. Through this mechanism, the generated description $D_i$ encapsulates the semantics from the node's history spanning from $S_1$ through $S_i$, effectively capturing its long-term, global temporal evolution.

In the above process, each interaction is fed into the LLM twice. Consequently, the overall complexity is limited to $O(|\mathcal{E}|)$, highlighting the efficiency of the explicit reasoning process.

## 4.3 INTEGRATING SEMANTICS AND GRAPH STRUCTURE

Next we present the component to integrate temporal semantics and dynamic graph structure features. For a node $v$ and a prediction time $t$, this module aims to obtain an integrated representation, which will be used for downstream tasks. As shown in Figure 2(c), we first divide the model into three components: Recent Semantic (RS) layer, Global Semantic (GS) layer, and graph structure layer. Let $L$ denote the total number of layers and $l$ denote the current layer index.

**Recent Semantic Layer.** Let $\mathcal{N}_v^t = \{I_i | t_i < t, I_i \in \mathcal{N}_v\}$. In Section 4.1, we have obtained the temporal feature $\mathbf{F}_i^{\text{rc}}$ for interaction $I_i \in \mathcal{N}_v^t$ and its corresponding timestamp $t_i$. Following (Xu et al., 2020), we use a learnable time encoder to encode the time interval $\Delta t_i = t - t_i$ into a $d_t$-dimensional vector to capture periodic temporal patterns:

$$\mathbf{R}_i^{(0)} = [\mathcal{P}\left(\mathbf{F}_i^{\text{rc}}\right) \ || \ \mathscr{T}(t - t_i)], \tag{3}$$

where $\mathcal{P}(\cdot) : \mathbb{R}^{d_{\text{LLM}}} \to \mathbb{R}^{d_t}$ is a projector, $[\cdot||\cdot]$ denotes concatenation, and $\mathscr{T}(\cdot)$ is the time encoder. Note that $d_t \ll d_{\text{LLM}}$ and thus the projection can also accelerate computing. Lastly, $\{\mathbf{R}_i^{(0)}\}$ are used as the input feature for the RS layer to compute the recent temporal semantics:

$$\mathbf{R}^{(l)} = \left[\mathbf{R}_0^{(l)}, \mathbf{R}_1^{(l)}, ..., \mathbf{R}_{|\mathcal{N}_v^t|}^{(l)}\right] = \text{RS\_Layer}\left(\left[\mathbf{R}_0^{(l-1)}, \mathbf{R}_1^{(l-1)}, ..., \mathbf{R}_{|\mathcal{N}_v^t|}^{(l-1)}\right]\right). \tag{4}$$

Given the strengths of Transformers in processing sequential data, we employ a Transformer Encoder to realize RS_Layer$(\cdot)$.

**Global Semantic Layer.** Similarly, we have obtained the global textual description $D_i$ for node $v$ in timestamp $\hat{t}_i$ in Section 4.2. We use a word embedding model (e.g., BERT) to transform each textual description $D_i$ into a hidden representation $\mathbf{F}_i^{\text{gb}} \in \mathbb{R}^{d_{\text{BERT}}}$ as global semantic features. To capture the dynamic changes in node features leading up to $t$, we then calculate the input features $\mathbf{G}_i^{(0)}$ as follows:

$$\mathbf{G}_i^{(0)} = \left[\mathcal{P}'(\mathbf{F}_i^{\text{gb}}) \ || \ \mathscr{T}(t - \hat{t}_i)\right], \tag{5}$$

where $\mathcal{P}'(\cdot) : \mathbb{R}^{d_{\text{Bert}}} \to \mathbb{R}^{d_T}$ is another projector. Lastly, the global feature are updated as:

$$\mathbf{G}^{(l)} = \left[\mathbf{G}_0^{(l)}, \mathbf{G}_1^{(l)}, ..., \mathbf{G}_{\hat{i}}^{(l)}\right] = \text{GS\_Layer}\left(\left[\mathbf{G}_0^{(l-1)}, \mathbf{G}_1^{(l-1)}, ..., \mathbf{G}_{\hat{i}}^{(l-1)}\right]\right). \tag{6}$$

where $\hat{i} = \max\{i|\hat{t}_i < t\}$ ensures that only features up to $t$ are considered to prevent future information leakage. Similar to the RS_layer, we use a Transformer Encoder to realize GS_Layer($\cdot$) in our implementation.

**Graph Structure Layer.** In this component, we first explicitly model dynamic graph structures by learning the structural representation $\mathbf{S}^{(l)}$. The graph structure layer is represented as:

$$\mathbf{S}^{(l)} = \text{Merge}^{\text{TGNN}} \left( \mathbf{M}^{(l-1)}, \text{MP}_{\mathcal{G}}(v, t) \right), \tag{7}$$

where $\text{MP}_{\mathcal{G}}(v, t)$ denotes the process of gathering and aggregating features from the neighbors of node $v$ in graph $\mathcal{G}$ up to time $t$, $\mathbf{M}^{(l-1)}$ is the integrated node feature, and $\text{Merge}^{\text{TGNN}}$ merges and updates $\mathbf{M}^{(l-1)}$ with $\text{MP}_{\mathcal{G}}(v, t)$. Benefiting from the modularity design of DyGRASP, our method is compatible with any existing message-passing temporal GNN. In our implementation, two representative temporal GNNs, DyGFormer (Yu et al., 2023) and TGAT (Xu et al., 2020), are used to demonstrate the compatibility of DyGRASP.

The integrated node feature $\mathbf{M}^{(l)}$ aims to fuse features from three components, i.e., recent semantics $\{\mathbf{R}_i^{(l)}\}$, global semantics $\{\mathbf{G}_i^{(l)}\}$), and graph structure $\mathbf{S}^{(l)}$, using Merge_Layer($\cdot$):

$$\mathbf{M}^{(l)} = \text{Merge\_Layer} \left( \mathcal{R}^r(\{\mathbf{R}_i^{(l)}\}), \mathcal{R}^g(\{\mathbf{G}_i^{(l)}\}), \mathbf{S}^{(l)} \right), \tag{8}$$

where $\mathbf{M}^{(l)} \in \mathbb{R}^{d_{\text{SF}}}$ represents the node's integrated feature, $\mathbf{M}^{(0)}$ is the node raw feature, $\mathcal{R}^r(\cdot)$ and $\mathcal{R}^g(\cdot)$ are readout functions responsible for summarizing the features from the RS and GS Layer, respectively. We set Merge_Layer as an MLP and $\mathcal{R}^r(\cdot)$ as a mean pooling operation:

$$\mathcal{R}^r(\{\mathbf{R}_i^{(l)}\}) = \text{Mean\_Pooling} \left( [\mathbf{R}_0^{(l)}, \mathbf{R}_1^{(l)}, \ldots] \right). \tag{9}$$

For $\mathcal{R}^g(\cdot)$, we simply take the last feature in the sequence, i.e.,

$$\mathcal{R}^g(\{\mathbf{G}_i^{(l)}\}) = \mathbf{G}_{\hat{i}}^{(l)}, \tag{10}$$

which is based on following considerations: (i) the Transformer Encoder in the GS_Layer($\cdot$) inherently aggregates information via the attention mechanism, potentially concentrating historical global semantics in the final feature $\mathbf{G}_{\hat{i}}^{(l)}$; (ii) the preceding global reasoning chain employs a RNN-like mechanism to propagate historical global semantic to the final feature. The final output of the model, $\mathbf{M}^{(L)}$, serves as the node's comprehensive feature representation for downstream tasks.

**Training for DyGRASP.** Following DTGB, We adopt an end-to-end training approach and supervise the model training using a binary cross-entropy loss for both the destination node retrieval and future link prediction tasks. Specifically, we use an MLP layer to take the final features (i.e., $\mathbf{M}_u^{(L)}$ and $\mathbf{M}_v^{(L)}$) to predict the probability score of a future interaction between nodes $u$ and $v$. For each true interaction $I_{u,v_{pos}}$ occurring at time $T$, we randomly sample a negative node $v_{neg}$ to create a false interaction $I_{u,v_{neg}}$, which did not actually occur. Finally, we use all the true and false samples for training.

## 5 EXPERIMENTS

In this section, we first show the superiority of DyGRASP over existing methods across various tasks and validate the generalizability of DyGRASP on multiple LLMs and base temporal GNNs. Then we separately validate the effectiveness of Recent/Global Semantic Reasoning and study the impact of the hyperparameter, as well as investigate the inference efficiency.

### 5.1 EXPERIMENTAL SETTINGS

**Evaluation Tasks and Datasets.** Following (Zhang et al., 2024), we evaluate models using the destination node retrieval task with Hit@10 metric under transductive and inductive settings and the future link prediction task with AP and ROC-AUC metrics. Except for main results, all other experiments use destination node retrieval as the evaluation task. We select four datasets from the

Table 1: Hit@10 (%) results for destination node retrieval. OOM means out of memory. We use **boldface** and underlining to denote the best and the second-best performance, respectively.

| Methods | Transductive | | | | Inductive | | | |
|---|---|---|---|---|---|---|---|---|
| | GDELT | Enron | Googlemap | Stack_elec | GDELT | Enron | Googlemap | Stack_elec |
| **JODIE** | 89.88±0.10 | 91.01±0.82 | OOM | OOM | 68.84±1.09 | 78.78±1.63 | OOM | OOM |
| **DyRep** | 85.34±0.61 | 85.79±2.81 | OOM | OOM | 61.63±4.01 | 68.53±1.68 | OOM | OOM |
| **CAWN** | 89.92±0.24 | 91.96±0.87 | 52.07±0.97 | 91.37±0.17 | 63.84±0.99 | 76.48±1.03 | 44.04±0.74 | 46.30±0.32 |
| **TCL** | 90.58±0.44 | 87.96±3.12 | 48.70±0.75 | 84.16±10.94 | 69.28±1.12 | 64.99±9.99 | 40.90±0.66 | 37.82±11.58 |
| **GraphMixer** | 88.67±0.05 | 87.36±1.07 | 48.34±0.23 | 93.25±0.04 | 63.92±0.12 | 66.83±1.08 | 40.71±0.19 | 52.04±0.34 |
| **Llama-3.1** | 41.01 | 92.09 | 53.97 | 34.24 | 31.75 | 79.55 | 43.56 | 33.59 |
| **TGAT** | 89.97±0.03 | 89.12±0.21 | 63.53±0.69 | 93.74±0.11 | 65.76±0.46 | 70.93±0.85 | 56.61±0.51 | 54.29±0.47 |
| **DyGRASP**(TGAT) | 91.96±0.17 | 96.25±1.11 | 82.79±0.17 | **99.74±0.04** | 71.78±0.28 | 86.76±2.07 | 78.63±0.20 | **99.77±0.09** |
| *Improv.* ↑ | *+1.99* | *+7.13* | *+19.26* | *+6.00* | *+6.02* | *+15.83* | *+22.02* | *+45.48* |
| **DyGFormer** | 91.64±0.24 | 92.04±1.7 | 51.32±2.13 | 94.39±0.12 | 72.49±0.37 | 82.52±2.34 | 43.84±1.92 | 56.23±0.52 |
| **DyGRASP**(DyGFormer) | **93.24±0.13** | **99.40±0.29** | **85.88±2.64** | 99.59±0.25 | **75.93±0.16** | **95.18±0.58** | **81.14±2.12** | 99.74±0.14 |
| *Improv.* ↑ | *+1.60* | *+7.36* | *+34.56* | *+5.20* | *+3.44* | *+12.66* | *+37.30* | *+43.51* |

DTGB benchmark (Zhang et al., 2024): GDELT, Enron, Googlemap, and Stack_elec, which cover diverse graph scales and domains.

**Model Configurations and Baselines.** We choose seven state-of-the-art temporal GNNs as baselines: JODIE (Kumar et al., 2019), DyRep (Trivedi et al., 2019), CAWN (Wang et al., 2021c), TCL (Wang et al., 2021a), GraphMixer (Cong et al., 2023), TGAT (Xu et al., 2020), and DyGFormer (Yu et al., 2023). We also evaluate the performance of Llama-3.1-8B-Instruct (Grattafiori et al., 2024), the primary LLM used in DyGRASP, as another baseline. For DyGRASP, we primarily utilize DyGFormer as the base temporal GNN and include TGAT to validate the generalizability. We mainly employ Llama-3.1-8B / Llama-3.1-8B-Instruct for implicit and explicit reasoning, respectively. More details can be found in Appendix E.

## 5.2 MAIN RESULTS

Table 2: Results for future link prediction. OOM means out of memory. We use **boldface** and underlining to denote the best and the second-best performance, respectively.

| Metric | Methods | Transductive | | | | Inductive | | | |
|---|---|---|---|---|---|---|---|---|---|
| | | GDELT | Enron | Googlemap | Stack_elec | GDELT | Enron | Googlemap | Stack_elec |
| | **JODIE** | 95.56±0.08 | 96.70±0.34 | OOM | OOM | 87.94±0.47 | 90.26±0.58 | OOM | OOM |
| | **DyRep** | 94.55±0.19 | 94.99±0.93 | OOM | OOM | 85.02±1.59 | 85.84±1.11 | OOM | OOM |
| | **CAWN** | 96.57±0.05 | 97.85±0.13 | 82.78±0.50 | 95.55±0.08 | 88.65±0.29 | 92.91±0.38 | 78.67±0.63 | 80.48±0.21 |
| | **TCL** | 96.63±0.14 | 96.31±1.44 | 80.71±0.40 | 89.58±9.50 | 89.95±0.38 | 88.27±4.37 | 76.13±0.46 | 73.09±10.44 |
| *AP* | **GraphMixer** | 95.95±0.05 | 96.01±0.22 | 80.60±0.05 | 96.17±0.02 | 87.30±0.06 | 87.60±0.30 | 76.13±0.18 | 82.75±0.20 |
| | **TGAT** | 96.41±0.02 | 96.78±0.04 | 87.71±0.27 | 96.59±0.10 | 88.30±0.14 | 89.54±0.33 | 84.97±0.23 | 84.24±0.51 |
| | **DyGFormer** | 97.02±0.03 | 97.86±0.01 | 81.63±0.40 | 96.70±0.10 | 91.09±0.15 | 93.60±0.22 | 77.23±0.72 | 84.67±0.34 |
| | **Llama-3.1** | 82.23 | 86.13 | 67.80 | 60.35 | 73.61 | 74.34 | 64.08 | 58.93 |
| | **DyGRASP**(TGAT) | **97.78±1.46** | 98.04±0.61 | **96.23±2.46** | 98.25±0.13 | **93.05±4.43** | 91.66±3.26 | **95.31±3.01** | 88.11±0.31 |
| | **DyGRASP**(DyGFormer) | 97.40±0.03 | **98.81±0.14** | 94.85±0.22 | **98.42±0.02** | 92.23±0.02 | **95.03±0.60** | 93.68±0.20 | **88.50±0.20** |
| | **JODIE** | 96.26±0.02 | 97.01±0.25 | OOM | OOM | 88.03±0.27 | 90.21±0.52 | OOM | OOM |
| | **DyRep** | 95.23±0.11 | 95.47±0.92 | OOM | OOM | 85.49±0.69 | 86.24±1.03 | OOM | OOM |
| | **CAWN** | 96.69±0.05 | 97.80±0.17 | 83.45±0.50 | 96.38±0.10 | 88.32±0.29 | 91.93±0.07 | 78.92±0.67 | 80.44±0.28 |
| | **TCL** | 96.77±0.13 | 96.54±1.24 | 80.90±0.41 | 89.23±11.74 | 89.56±0.38 | 88.08±4.15 | 75.45±0.66 | 73.16±10.59 |
| *AUC* | **GraphMixer** | 96.14±0.02 | 96.13±0.21 | 80.84±0.07 | 96.86±0.02 | 87.19±0.13 | 87.33±0.26 | 75.21±0.25 | 83.25±0.13 |
| | **TGAT** | 96.58±0.02 | 96.92±0.04 | 88.19±0.33 | 97.19±0.06 | 88.33±0.12 | 89.46±0.34 | 85.34±0.30 | 84.56±0.29 |
| | **DyGFormer** | 97.10±0.03 | 97.69±0.03 | 81.91±0.37 | 97.31±0.07 | 90.59±0.15 | 92.68±0.28 | 76.34±0.78 | 85.24±0.31 |
| | **Llama-3.1** | 83.31 | 86.86 | 69.38 | 55.41 | 74.03 | 78.26 | 66.17 | 56.04 |
| | **DyGRASP**(TGAT) | 97.22±0.04 | 98.34±0.39 | 94.87±0.04 | 98.79±0.06 | 90.74±0.10 | 92.38±2.33 | 93.47±0.08 | 85.91±0.39 |
| | **DyGRASP**(DyGFormer) | **97.60±0.04** | **98.86±0.14** | **94.89±0.26** | **98.87±0.01** | **92.19±0.06** | **94.85±0.63** | **93.58±0.25** | **86.69±0.20** |

**DyGRASP consistently outperforms all existing methods with substantial improvements.** As shown in Table 1 we can observe that under both transductive and inductive settings, Dy-GRASP(DyGFormer) outperforms existing methods on all datasets, including traditional temporal GNNs and Llama-3.1-8B-Instruct, demonstrating the superiority of DyGRASP. Notably, benefiting from the strong generalization of LLMs on semantic understanding, DyGRASP achieves higher improvements in the inductive setting compared to the transductive setting. Moreover, as shown in Table 1 and Figure 8, the performance improvement over the base temporal GNN is more pronounced on datasets with richer textual information, highlighting the important role of the LLM in our method.

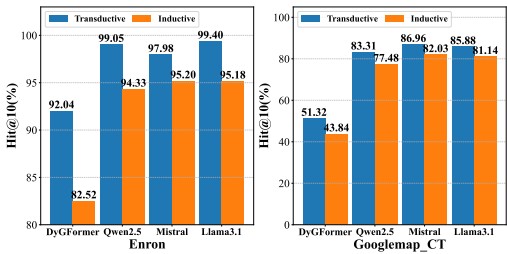 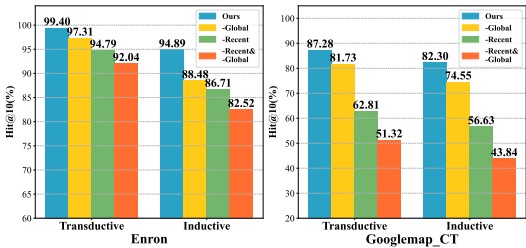

Figure 3: Results with different LLM backbones. Our method is compatible with various LLMs.

Figure 4: Results of ablation studies. Both global and recent modules contribute to our method.

**DyGRASP demonstrates generalizability across both base temporal GNNs and LLMs.** As shown in Table 1 , the results indicate that DyGRASP consistently yields significant performance improvements compared to the original temporal GNNs when either using TGAT or DyGFormer as the base temporal GNN. Furthermore, we validate the effectiveness of DyGRASP with three different families of LLMs, including Qwen2.5, Mistral, and Llama-3.1, on the Enron and Googlemap datasets. As shown in Figure 3, DyGRASP consistently outperforms the base temporal GNN regardless of the specific LLM and exhibits minimal variation across different LLMs, demonstrating the exceptional general applicability and compatibility of DyGRASP.

## 5.3 ABLATION STUDIES AND ANALYSES

**Ablation studies.** To validate the components of our method, we evaluate the following ablations: (i) removing the global semantic layer (denoted as -Global); (ii) removing the recent semantic layer (denoted as -Recent); (iii) removing both layers (denoted as -Recent&-Global).

As illustrated in Figure 4, under both the transductive and inductive settings, adding either the recent or global temporal semantic reasoning module individually leads to improvements in performance. Furthermore, the performance obtained with both modules added simultaneously surpassed those achieved when only a single module is incorporated, which strongly confirms the complementarity of recent-global features and the effectiveness of both LLM reasoning designs.

**Hyperparameter Sensitivity Analysis.** Next, we investigate the key hyperparameters of DyGRASP, including the number of global segments $s$ in global semantic reasoning and the maximum interaction length $c$ in recent temporal semantic reasoning. We separately study the effects of $c$ and $s$ using the features given by either the recent semantic layer or the global semantic layer. We report results on the Googlemap dataset, while other datasets show similar trends.

*Number of global segments $s$.* As shown in Figure 5, model performance consistently improves as $s$ increases. We hypothesize that it is because a larger number of segments leads to more frequent reasoning for the LLM and smaller intervals between two reasoning time points, which allows the LLM to give smoother global temporal semantic variations. However, a higher number of $s$ also leads to a proportional increase in the inference time.

*Maximum interaction length $c$.* As illustrated in Figure 5, model performance initially improves significantly as $c$ increases but then decreases. A plausible reason is that, compared to independent encoding of text attribute ($c = 0$), the recent semantic reasoning captures temporal semantic dependencies between interactions. A larger $c$ allows observation of more historical interactions, enriching the recent temporal semantics thus leading to performance gains. However, when $c$ sur-

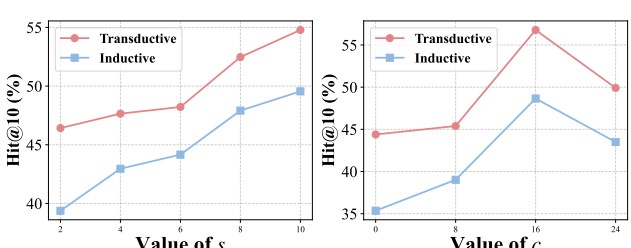

Figure 5: Results of sensitivity analysis.

passes 16, performance starts to decline plausibly because of increased input length, i.e., the perfor-

mance of LLMs can degrade significantly as input length increases (Wang et al., 2024a; Levy et al., 2024).

**Efficiency Analysis.** Due to the decoupling of LLM inference from the online prediction task, the inference of actual prediction task can be as fast as a lightweight temporal GNN. Furthermore, the computationally intensive LLM reasoning is an offline, one-time pre-processing step to extract temporal semantic features. These features are then stored (e.g., in a vector database) for fast retrieval. Here we first investigate the inference efficiency of LLMs in our method and then provide a speed comparison on online prediction task. The results are shown in Table 3 and Table 4 where the inference time of LLMs is measured by Nvidia-A100 GPU-Days.

Table 3: Speed comparison (edges/second) on online prediction task.

| Dataset | JODIE | DyRep | CAWN | TCL | GraphMixer | TGAT | DyGFormer | Llama-3.1 | **DyGRASP (Ours)** |
|---|---|---|---|---|---|---|---|---|---|
| Enron | 3,372 | 2,872 | 313 | 1,157 | 2,410 | 587 | 1,077 | 15 | **729** |
| Googlemap_CT | OOM | OOM | 307 | 2,621 | 3,098 | 545 | 1,165 | 13 | **757** |

*Online prediction efficiency.* Despite incorporating an LLM, DyGRASP's inference speed is competitive with other TGNNs and is notably faster than complex attention-based (e.g., TGAT) and random-walk-based (e.g., CAWN) models. This speed is orders of magnitude faster than that of a pure LLM baseline (e.g., Llama 3.1), demonstrating the high efficiency of our method at online prediction.

Table 4: Token consumption and inference time of different components.

| Reasoning configuration | Token Consumption | | | | Inference time | | | |
|---|---|---|---|---|---|---|---|---|
| | GDELT | Enron | Googlemap | Stack_elec | GDELT | Enron | Googlemap | Stack_elec |
| Recent w/o node-centric reasoning | 1.68B | 3.62B | 2.42B | 7.00B | 2.37 | 4.34 | 2.45 | 7.21 |
| Recent w/ node-centric reasoning | 0.10B | 0.86B | 0.35B | 0.96B | 0.16 | 0.84 | 0.52 | 1.21 |
| Global Reasoning | 0.03B | 0.25B | 0.52B | 0.48B | 0.04 | 0.37 | 0.89 | 1.42 |

*Recent semantic reasoning efficiency.* As discussed in Section 4.1, we introduce node-centric implicit reasoning to address the inference efficiency. The results show that the node-centric reasoning method significantly reduces the reasoning time across all datasets, which is consistent with our theoretical analysis. This efficiency gain stems from a critical trade-off design: by shifting from an edge-centric to a node-centric view, we reduce the theoretical complexity from a prohibitive $O(|\mathcal{E}| \times d)$ to a manageable $O(|\mathcal{E}|)$. Furthermore, the sliding window size $c$ acts as a control lever for this trade-off. We select $c$ to maximize the capture of historical context while avoiding the computational burden and performance degradation associated with excessively long LLM inputs.

*Global semantic reasoning efficiency.* We also report the inference time in Table 4. We could observe that the time consumption for this module is also low, highlighting the efficiency of DyGRASP. This performance relies on the granularity trade-off governed by the number of segments $s$. Instead of processing a node's entire vast history at once, we partition it into $s$ segments. While a larger $s$ provides more fine-grained snapshots of semantic evolution, it leads to a linear increase in inference time. Our design balances this trade-off to ensure computational feasibility while effectively capturing long-term global dynamics.

## 6 CONCLUSION

In this paper, we propose DyGRASP, an efficient and effective model for DyTAGs by capturing the recent-global temporal semantics. We leverage the implicit and explicit reasoning capabilities of LLMs to uncover the recent and global semantics in DyTAGs, respectively, and integrate these temporal semantic features with dynamic graph structural features through a temporal GNN. Experiments show that DyGRASP achieves significant improvement of up to 34% in the Hit@10 metric for the destination node retrieval task compared to state-of-the-art methods.

## 7 ACKNOWLEDGMENT

This work was supported in part by the National Natural Science Foundation of China (No. 92570113, 62472018).

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

## A LIMITATIONS

One limitation of our paper is that we only test our method on DTGB benchmarks, considering that it is the only publicly available benchmark for DyTAGs. Experiments on more diverse real-world datasets and tasks will further demonstrate the usefulness of DyGRASP. Another direction worthy of exploration is extending our method into more complicated graph types, such as DyTAGs with heterogeneous nodes and edges or hyper-relations.

## B NOTATIONS

In this section, we provide important notations used in this paper, detailed in Table 5.

Table 5: Notations and descriptions.

| Notation | Description |
|---|---|
| $\mathcal{G}$ | A dynamic text-attributed graph. |
| $\mathcal{E}$ | The edge set of the DyTAG. |
| $\mathcal{T}$ | The timestamp set of the DyTAG. |
| $\mathcal{D}$ | The node text attribute set of the DyTAG. |
| $\mathcal{R}$ | The edge text attribute set of the DyTAG. |
| $\mathcal{N}_v$ | The chronological list of interactions with node $v$'s first-hop neighbors. |
| $\mathcal{G}_t$ | The state of the DyTAG before timestamp $t$. |
| $I_{u,v}$ | An interaction between two nodes. |
| $\mathbf{F}_i^{\text{rc}}$ | The raw recent temporal semantic feature of an interaction $I_i$. |
| $S^i$ | A segment of interactions, which contains interactions occurring between two partition timestamps. |
| $\mathbf{T}^{\text{gb}}$ | The set of all partition timestamps. |
| $D_i$ | The description of a node before a certain timestamp $S_i$. |
| $\mathbf{F}_i^{\text{gb}}$ | The raw global temporal semantic feature of a node before the $i^{th}$ partition timestamp. |
| $\mathbf{R}^{(l)}$ | The processed recent temporal semantic feature of the $l$-th layer. |
| $\mathbf{G}^{(l)}$ | The processed global temporal semantic feature of the $l$-th layer. |
| $\mathbf{S}^{(l)}$ | The graph structure feature given by a temporal GNN of the $l$-th layer |
| $\mathcal{R}^r(\cdot)$ | A readout function to get the recent semantic feature of a certain node from a list of $\mathbf{R}^{(l)}$ |
| $\mathcal{R}^g(\cdot)$ | A readout function to get the global semantic feature of a certain node from a list of $\mathbf{G}^{(l)}$ |

## C PROOF OF REASONING COMPLEXITY

In this section, we give a complexity proof of recent semantic reasoning in Section 4.1 (i.e., the number of consumed input tokens), including both intuitive approach without node-centric reasoning

and the proposed method with node-centric reasoning. For simplicity of analysis, we assume that the degree of each node is $d$ and the interaction of each node with its $d$ neighbor nodes occurs at distinct times $t_i(1 \le i \le d)$, where $t_i = t_j$ if and only if $i = j$. In this case, we have $O(|\mathcal{E}|) = O(|\mathcal{V}| \times d)$.

**Complexity of intuitive approach without node-centric reasoning.** Consider node $u$ and its historical interaction sequence:

$$\mathcal{N}_u = \{I_1^u, I_2^u, ..., I_d^u\}, \text{where } t_1 \le t_2 \le ... \le t_d. \tag{11}$$

For a historical interaction $I_i^u$, assume its interacted node is $v$. To provide temporal context for $I_i^u$, the input content to the LLM is:

$$\text{Input} = \{I_j^k | k \in \{u, v\}, 1 \le j < i\} \cup \{I_i^u\}, \tag{12}$$

which totals $2 \times (i - 1) + 1$ interactions. Therefore, the input complexity for a single node is:

$$O(\sum_{i=1}^{d}(2i - 1)) = O(d^2). \tag{13}$$

Since each edge belongs to two nodes, the total input complexity for all nodes is:

$$O(\frac{|\mathcal{V}|}{2}) \times O(d^2) = O(|\mathcal{E}| \times d). \tag{14}$$

**Complexity of our proposed method with node-centric reasoning.** Consider node $u$ and its historical interaction sequence $\mathcal{N}_u$, as shown in Eq. 11. This sequence is divided by a sliding window into $\lceil \frac{d}{\frac{c}{2}} \rceil - 1$ input batches, where $c$ is the window length. The complexity of each batch is $O(c)$. Therefore, the input complexity for node $u$ is:

$$O((\lceil \frac{d}{\frac{c}{2}} \rceil - 1) \times c) = O(d). \tag{15}$$

The final input complexity for all nodes is $O(d \times |\mathcal{V}|) = O(|\mathcal{E}|)$.

## D PROMPT FOR LLM REASONING

In this section, we provide dataset-specific prompt templates used for LLM reasoning, including implicit reasoning for recent temporal semantics and explicit reasoning for global temporal semantics. We select prompt for Googlemap as an example. The remaining dataset follows a similar structure.

> Below is a series of reviews on business entities given by {user_description}, an user of Google map. These reviews are arranged in chronological order from earliest to latest. Capture the potential relationships between them.
>
> Timestamp: {timestamp A} | Business Description: '{business description A}' | Review: '{review_content A}'
> Timestamp: {timestamp B} | Business Description: '{business description B}' | Review: '{review_content B}'
> Timestamp: {timestamp C} | Business Description: '{business description C}' | Review: '{review_content C}'
> ...

Figure 6: The prompt used for Googlemap of recent semantic reasoning.

## E DETAILED EXPERIMENTAL SETTINGS

**Model Configurations.** For DyGRASP, we primarily utilize DyGFormer as the base temporal GNN and include TGAT to validate the generalizability. We mainly employ Llama-3.1-8B / Llama-3.1-8B-Instruct for implicit and explicit reasoning, respectively. We also include Qwen2.5-7B / Qwen2.5-7B-Instruct (Yang et al., 2024) and Mistral-7B / Mistral-7B-Instruct (Jiang et al., 2023) to verify the generalizability of DyGRASP across different LLMs. We set $c$ to 64 for GDELT and 16 for the remaining datasets. $c$ is selected based on following considerations:

- **Data Characteristics:** $c$ is adapted to the average node degree. Datasets with denser histories, like GDELT, require a larger $c$ to capture sufficient temporal context.

```
### Instruction
 Here is information about an user of Google map and its description at timestamp {last_describe_time}. Taking
into account its recent reviews on business entities, please provide a new summative description of this user at
timestamp {current_describe_time} based on your understanding. Return in the format "Description:<Your
Description>".
### User Information
 {raw_user_description}
### Description
 {last_user_description}
### Recent Reviews
Timestamp: {timestamp A} | Business Description: '{business description A}' | Review: '{review_content A}'
Timestamp: {timestamp B} | Business Description: '{business description B}' | Review: '{review_content B}'
Timestamp: {timestamp C} | Business Description: '{business description C}' | Review: '{review_content C}'
...
```

Figure 7: The prompt used for Googlemap of global semantic reasoning.

- **Computational & LLM Constraints:** A larger $c$ increases input sequence length, posing challenges for computational resources and the LLM's hard token limit. For datasets with lengthy text attributes (e.g., Enron and Stack_elec), $c$ is carefully bounded by these resource constraints.
- **Model Performance:** Crucially, LLM performance degrades on excessively long inputs, especially as they approach or exceed the model's context window—a phenomenon we analyze in Figure 5. This is the most critical interaction with token limits. We select $c$ to maximize information gain while avoiding this performance cliff.

We set $s$ to 8 across all datasets. The impact of these two hyperparameters is investigated in Section 5.3. The total number of layers $L$ is set to 2.

**Implementation Details.** We adopt the experimental setup provided by DTGB benchmark. All datasets are split into training, validation, and test sets with a ratio of 0.7:0.15:0.15. All models are trained using the Adam (Kingma & Ba, 2014) optimizer with a batch size of 256 and a learning rate of 0.0001. Training runs for up to 50 epochs, with validation performed every 5 epochs. We employ an early stopping mechanism with a patience of 5 epochs. Model-specific hyperparameters for each baseline are set according to the optimal values recommended by the DTGB benchmark (see Table 7 for details). Text attributes within the datasets, as well as the text generated by the LLM during explicit reasoning, are vectorized with the bert-base-uncased model (Devlin et al., 2019). All training was conducted on NVIDIA A100 40G GPUs.

### E.1 DESCRIPTIONS OF DATASETS

We use four datasets for the experiments, including GDELT, Enron, Googlemap and Stack_elec. We provide the statistics of datasets in Table 6. We also summarize the text attribute length of these datasets in Figure 8.

Table 6: Statistics of datasets.

| Dataset | Nodes | Edges | Domain | Timestamps | Bipartite Graph |
|---|---|---|---|---|---|
| **GDELT** | 6,786 | 1,339,245 | Knowledge graph | 2,591 | × |
| **Enron** | 42,711 | 797,907 | E-mail | 1,006 | × |
| **Googlemap** | 111,168 | 1,380,623 | E-commerce | 55,521 | ✓ |
| **Stack_elec** | 397,702 | 1,262,225 | Multi-round dialogue | 5,224 | ✓ |

We provide descriptions of these datasets as follows.

- **GDELT**: this dataset comes from the "Global Database of Events, Language, and Tone" project, which aims to build a catalog of political behavior covering countries worldwide. In this dataset, nodes represent political entities and the node text attribute is the entity name while edges represent political relationships or actions between entities and the edge text attribute represents the description of the behavior.

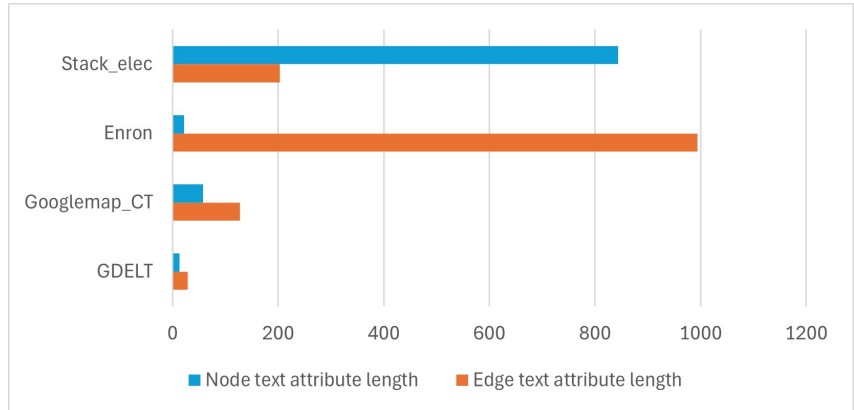

Figure 8: Text attribute length of used DyTAG datasets.

- **Enron**: this dataset originates from the email communications of employees at the Enron energy company. Here, nodes represent employees and the node text attribute is the employee's email address while edges represent the sending of emails and the edge text attribute is the email content.

- **Googlemap**: this dataset is extracted from the Connecticut State (CT) portion of the "Google Local Data" project, containing business review information from Google Maps. In this dataset, nodes are users or businesses and the node text attribute is the user name or business description while edges represent reviews from users to businesses and the edge text attribute is the specific review content.

- **Stack_elec**: this dataset comes from anonymized data of user question-and-answer content related to electronics technology on the Stack Exchange website. In this dataset, nodes are questions or users on the site and the node text attribute is the question description or user profile while edges represent answers or comments by users on questions and the edge text attribute is the content of the answer or comment.

### E.2 DESCRIPTION OF BASELINES

In this section, we provide detailed descriptions of the baselines.

- **JODIE** (Kumar et al., 2019): JODIE is a coupled recurrent neural network model that learns dynamic embedding trajectories for users and items based on their temporal interactions. JODIE utilizes a novel projection operator to predict the future trajectory of embeddings and employs a scalable t-Batch training algorithm, significantly outperforming baseline methods in future interaction and user state change prediction tasks.

- **DyRep** (Trivedi et al., 2019): DyRep is a framework for learning representations on dynamic graphs by modeling topological evolution and node interactions as two distinct but coupled temporal point processes. It uses a temporal-attentive network to learn evolving node embeddings that capture the interplay between these structural and activity dynamics over continuous time.

- **CAWN** (Wang et al., 2021c): CAWN is a neural network model for inductive representation learning on temporal networks. CAWN captures network dynamics using temporal random walks and a novel anonymization strategy, outperforming prior methods in predicting future links, particularly on unseen parts of networks.

- **TCL** (Wang et al., 2021a): TCL is a Transformer-based dynamic graph modeling method, which optimizes dynamic node representations through contrastive learning. TCL adopts a two-stream encoder architecture to respectively process the temporal neighborhood information of the target interaction nodes and fuses them at the semantic level via a co-attentional Transformer.

- **GraphMixer** (Cong et al., 2023): GraphMixer is a conceptually simple architecture for temporal link prediction using only MLPs and neighbor mean-pooling, intentionally avoiding complex RNN and self-attention mechanisms. Despite its simplicity, GraphMixer achieves state-of-the-art

performance on benchmarks, demonstrating better results, faster convergence, and improved generalization compared to more complicated models.

- **TGAT** (Xu et al., 2020): TGAT utilizes a self-attention mechanism combined with a novel functional time encoding technique based on harmonic analysis. TGAT efficiently aggregates temporal-topological neighborhood information to generate time-aware embeddings for both existing and newly appearing nodes.

- **DyGFormer** (Yu et al., 2023): DyGFormer is a new Transformer-based architecture for dynamic graph learning that uses neighbor co-occurrence encoding and sequence patching to effectively capture node correlations and long-term dependencies from historical first-hop interactions.

### E.3 CONFIGURATIONS OF DIFFERENT METHODS

We present the hyperparameters for all baselines in Table 7, which are recommended by the official DTGB benchmark.

Table 7: Hyper-parameter setting for all baselines.

| Methods | # layers | # heads | dropout | time_feat_dim | channel_embedding_dim | patch_size | max_input_sequence_length |
|---|---|---|---|---|---|---|---|
| JODIE | 1 | 2 | 0.1 | 100 | / | / | / |
| DyRep | 1 | 2 | 0.1 | 100 | / | / | / |
| CAWN | / | / | 0.1 | 100 | / | / | / |
| TCL | 2 | 2 | 0.1 | 100 | / | / | / |
| GraphMixer | 2 | / | 0.1 | 100 | / | / | / |
| TGAT | 2 | 2 | 0.1 | 100 | / | / | / |
| DyGFormer | 2 | 2 | 0.1 | 100 | 50 | 1 | 48 |

| Methods | # depths | # walk_heads | walk_length | position_feat_dim | # neighbors | sample_neighbor_strategy | time_scaling_factor |
|---|---|---|---|---|---|---|---|
| JODIE | / | / | / | / | 10 | recent | / |
| DyRep | / | / | / | / | 10 | recent | / |
| CAWN | / | 8 | 1 | 768 | 32 | time_interval_aware | 1e-6 |
| TCL | 21 | / | / | / | 20 | recent | / |
| GraphMixer | / | / | / | / | 20 | recent | / |
| TGAT | / | / | / | / | 20 | recent | / |
| DyGFormer | / | / | / | / | / | recent | / |

### E.4 COMPARISON AGAINST EXISTING LLM-FOR-DYTAGS METHOD

Here we provide a comparison with LKD4DyTAG. Note that the source code of LKD4DyTAG is not available, so we directly compare with their reported results. The results are presented below.

Table 8: Comparison on future link prediction against LKD4DyTAG.

| Metric | Methods | Transductive | | | | Inductive | | | |
|---|---|---|---|---|---|---|---|---|---|
| | | GDELT | Enron | Googlemap_CT | Avg. | GDELT | Enron | Googlemap_CT | Avg. |
| *AP* | **LKD4DyTAG** | 97.76 | **99.43** | 91.44 | 96.21 | **93.69** | **95.50** | 88.90 | 92.70 |
| | **DyGRASP** (TGAT) | **97.78** | 98.04 | **96.23** | **97.35** | 93.05 | 91.66 | **95.31** | **93.34** |
| | **DyGRASP** (DyGFormer) | 97.40 | 98.81 | 94.85 | 97.02 | 92.23 | 95.03 | 93.68 | 93.65 |
| *AUC* | **LKD4DyTAG** | **99.98** | 98.87 | 91.27 | 96.71 | **95.77** | 93.67 | 88.57 | 92.67 |
| | **DyGRASP** (TGAT) | 97.22 | 98.34 | 94.87 | 96.81 | 90.74 | 92.38 | 93.47 | 92.20 |
| | **DyGRASP** (DyGFormer) | 97.60 | 98.86 | **94.89** | **97.12** | 92.19 | **94.85** | **93.58** | **93.54** |

As shown in Table 8, on the average results across three datasets, DyGRASP consistently outperforms LKD4DyTAG for both AP and AUC under both transductive and inductive settings. A more detailed analysis reveals the specific strengths of our method:

- On datasets with less textual information like GDELT, where graph structure is more critical, LKD4DyTAG performs slightly better. This is likely due to its custom-designed base TGNN being more effective at capturing structural features.

- However, on text-rich datasets with complex semantic patterns like Googlemap_CT, DyGRASP demonstrates a significant lead. This validates our core contribution: DyGRASP's explicit design for modeling recent and global temporal semantics provides superior semantic reasoning capabilities compared to existing methods.

## F    Case Study

We provide a case study from the Googlemap_CT dataset below, which demonstrates how the model uses global summarization to predict whether customer **Joe Golin** will visit the business **Liquor Depot** at timestamp 48264.

---

**Global Summarization for Customer Node "Joe Golin" (at $t = 47960$)**

**Description:** Joe Golin is a customer *who values exceptional service, high-quality food, and unique dining experiences*, particularly in Asian cuisine, seafood restaurants, and *pubs*. His enthusiasm for crab ragoons remains unwavering, with his positive reviews at Toshi Izakaya, J Timothy's Taverne, and 707 Asian Fusion showcasing his love for this dish. He is also a fan of imaginative waffles and great pizza, as evident from his positive reviews of THE SLOPPY WAFFLE LLC and Pizzeria Marzano. While he is willing to try new restaurants, his reviews suggest that he has high expectations and can be disappointed if they do not meet his standards, as seen in his review of Taco Bell. Joe also values excellent service, as evident from his positive reviews of Black Diamond Body Piercing, Mulligan's, O'Connor's Public House, and Onion Mountain Kitchen. His recent visits to Dos Amigos and Midas, where he experienced excellent service, further emphasize his appreciation for friendly and efficient service.

---

**Global Summarization for Business Node "Liquor Depot" (at $t = 42596$)**

**Description:** Liquor Depot, Inc. is a highly-recommended liquor store offering an extensive selection of beers, wines, and spirits at competitive prices. *Their knowledgeable staff provides excellent assistance in finding the perfect products, frequently featuring a wide range of bargain wines and an exceptional bourbon selection, including rare and hard-to-find options.* The store continues to host events, such as the "Bourbon Bash," allowing customers to purchase limited-edition bottles. **They boast an impressive selection of 12 packs, wine, and whiskey, catering to various tastes and preferences**. The store is praised for its great prices, *friendly staff*, and beautiful store environment, making it a go-to destination for liquor enthusiasts in Connecticut. The store's ability to adapt to changing tastes and trends, such as great St. Patrick's displays, further solidifies its reputation as a top choice.

---

By processing these global summaries, our model can identify strong semantic alignments between the user's preferences and the business's offerings:

- **Interest Alignment:** Joe Golin's summarized interest in "pubs" directly corresponds with "Liquor Depot" being a liquor store.
- **Service Quality Alignment:** Joe's high value for "exceptional service" is matched by Liquor Depot's reputation for "knowledgeable" and "friendly staff" who provide "excellent assistance".
- **Unique Experience Alignment:** Joe's preference for "high-quality food and unique dining experiences" aligns with Liquor Depot's "exceptional bourbon selection," "rare and hard-to-find options," and "impressive selection," which constitute a high-quality and unique product experience.

Based on this semantic reasoning, the model can infer a high probability that Liquor Depot meets Joe Golin's criteria, thus predicting his visit at the future timestamp. This illustrates how global summarization provides crucial, high-level context that goes beyond individual interactions to improve predictive accuracy.

## G    Use of LLM assistants

We used a LLM for the sole purpose of copyediting the paper to improve its clarity and readability.

