# OpenReview forum: "Global-Recent Semantic Reasoning on Dynamic Text-Attributed Graphs with Large Language Models"
_ICLR.cc/2026/Conference — ICLR 2026 Poster_

### Official Review · Reviewer_jwom · 2025-10-20

**Soundness:** 2
**Presentation:** 3
**Contribution:** 3
**Rating:** 6
**Confidence:** 4

**Summary:**

The paper addresses the challenge of Dynamic Text-Attribute Graphs (DyTAGs), which feature time-evolving graph interactions and associated text attributes that are common in real-world applications. Existing methods, like Graph Neural Networks (GNNs) and Large Language Models (LLMs), mainly focus on static Text-Attribute Graphs (TAGs). Extending these approaches to DyTAGs is difficult because they fail to account for recent-global temporal semantics, which include both the recent semantic dependencies among interaction texts and the global semantic evolution of nodes over time. Additionally, applying LLMs to the vast and evolving text data in DyTAGs introduces efficiency problems.

To solve these challenges, the authors propose a novel method called Dynamic Global-Recent Adaptive Semantic Processing (DyGRASP). DyGRASP combines LLMs and temporal GNNs to reason on DyTAGs in an efficient and effective manner.

**Strengths:**

1. DyGRASP effectively combines both recent and global temporal semantics by utilizing sliding window mechanisms for short-term dependencies and RNN-like structures for long-term dependencies. This dual approach improves the model's ability to capture evolving patterns in DyTAGs.

2. The combination of LLMs for semantic processing and temporal GNNs for structural reasoning is an effective solution for addressing both the textual and graph-based challenges in DyTAGs. The model’s design optimizes the use of LLMs to handle the evolving text efficiently, solving a major bottleneck in current systems. The idea of implicit and explicit reasoning paths may provide some insights to the researchers in this field.

3. This paper provides a comprehensive experimental result to verify its effectiveness.

**Weaknesses:**

1. While the integration of LLMs and temporal GNNs is innovative, it also increases the overall complexity of the model. This dual integration might result in higher computational costs and more difficult tuning for large-scale applications.

2. The paper highlights the efficiency of DyGRASP in handling evolving texts, but it would benefit from a more in-depth discussion of the computational trade-offs involved in its design, particularly when applied to large-scale DyTAGs with vast amounts of dynamic text.

**Questions:**

Authors may consider providing more analysis about the efficiency of the framework.

---

> ### Author Response · Authors · 2025-11-17
> **Authors' Response**
>
> > *W1. While the integration of LLMs and temporal GNNs is innovative, it also increases the overall complexity of the model. This dual integration might result in higher computational costs and more difficult tuning for large-scale applications.*
>
> **A1**. Thank you for this insightful comment. We agree that integrating LLMs, with their vast number of parameters, inevitably introduces additional computational costs. We have paid close attention to this issue. To address this, we have the following key design:
>
> Firstly, we proposed an efficient reasoning design that reduces the inference complexity to $O(|\mathcal{E}|)$, minimizing the extra computational burden. As empirically demonstrated in Table 3, our method leads to a multi-fold decrease in inference time compared to a vanilla implementation, greatly reducing the cost for large-scale applications. We believe this manageable overhead is a valuable trade-off for the significant performance improvement. Our model, DyGRASP, achieves up to a 34% gain on the Hit@10 metric, a substantial leap that justifies the inclusion of LLM-based reasoning.
>
> Furthermore, as we discussed in the "Efficiency Analysis" section (Lines 458-462), DyGRASP decouples the LLM reasoning from the online prediction task, making the inference phase as fast as a lightweight TGNN. Here we present the inference speed (edges/second) comparison to validate this point:
>
> | Inference Speed (edges/s) | Enron   | Googlemap_CT |
> | ------------------------- | ------- | ------------ |
> | JODIE                     | 3,372   | OOM          |
> | DyRep                     | 2,872   | OOM          |
> | CAWN                      | 313     | 307          |
> | TCL                       | 1,157   | 2,621        |
> | GraphMixer                | 2,410   | 3,098        |
> | TGAT                      | 587     | 545          |
> | DyGFormer                 | 1,077   | 1,165        |
> | Llama-3.1                 | 15      | 13           |
> | **DyGRASP (Ours)**        | **729** | **757**      |
>
> Despite incorporating an LLM, DyGRASP's inference speed is competitive with other TGNNs and is notably faster than complex attention-based (e.g., TGAT) and random-walk-based (e.g., CAWN) models. This speed is orders of magnitude faster than that of a pure LLM baseline (e.g., Llama 3.1), demonstrating the high efficiency of our method at inference time. We will add this comparison to our revised paper.
>
> > *W2. The paper highlights the efficiency of DyGRASP in handling evolving texts, but it would benefit from a more in-depth discussion of the computational trade-offs involved in its design, particularly when applied to large-scale DyTAGs with vast amounts of dynamic text.*
>
> **A2**. Thank you for your insightful feedback. Here we provide a more in-depth discussion of the computational trade-offs involved in DyGRASP.
>
> One of our core design philosophies for DyGRASP is to balance the powerful semantic reasoning of LLMs with the computational realities of large-scale dynamic graphs. This is achieved through several deliberate trade-offs:
>
> **Offline Pre-processing vs. Online Inference.** The most significant design choice is decoupling the computationally intensive LLM reasoning into a one-time, offline pre-processing step. As analyzed in Section 5.3, this allows us to generate rich semantic features that can be stored and retrieved efficiently. The online prediction task then relies only on the lightweight and fast temporal GNN. This makes DyGRASP highly scalable for real-world applications where inference speed is critical.
>
> **Node-Centric Reasoning Trade-off.** As proven in Appendix C, our node-centric approach reduces the complexity of capturing recent semantics from a prohibitive $O(|\mathcal{E}|*d)$ (edge-centric) to a much more manageable $O(|\mathcal{E}|)$. The key trade-off here could also be governed by the sliding window size $c$.
>
> **Global Reasoning Granularity Trade-off.** For global semantics, we partition a node's history into $s$ segments.
>
> * **Benefit:** A larger $s$ provides more fine-grained snapshots of a node's semantic evolution.
>
> * **Cost:** The inference time for this module scales linearly with $s$.
>
>   This design avoids feeding an entire, potentially vast, interaction history to the LLM at once, trading a small degree of temporal precision for a massive gain in feasibility and efficiency.
>
> In summary, DyGRASP’s efficiency stems from its modular design, which strategically uses LLMs in an offline, highly optimized manner while relying on efficient GNNs for online tasks. The hyperparameters $c$ and $s$ offer explicit controls to manage the trade-off between semantic richness and computational cost. We will add a dedicated subsection to elaborate on these trade-offs in the final version.

---

> > ### Comment · Reviewer_jwom · 2025-11-24
> >
> > Thanks for your reply and it partly addresses my concerns. Given the current review is already a positive score, I would like to keep my score.

---

> > > ### Author Response · Authors · 2025-11-25
> > > **Response**
> > >
> > > Dear Reviewer,
> > >
> > >
> > > Thank you for your insightful comments and positive evaluation. Please feel free to share any additional feedback you may have.
> > >
> > >
> > > Best regards,
> > > Authors

---

### Official Review · Reviewer_amvQ · 2025-10-26

**Soundness:** 3
**Presentation:** 3
**Contribution:** 3
**Rating:** 8
**Confidence:** 4

**Summary:**

This paper proposes an innovative framework named DyGRASP for the semantic understanding and reasoning problems of dynamic text attribute graphs. Targeting the insufficient semantic understanding ability of traditional temporal graph neural networks and the low efficiency of large language models, the authors propose a sliding window mechanism to efficiently capture recent temporal semantics, and use LLM to summarize the chain-structure long-range semantics of nodes for global information modeling. Experimental results on 4 benchmark datasets showcase its performance superiority and the generalization ability to various LLMs and temporal GNNs.

**Strengths:**

S1. The paper is well-written, with clear and engaging prose. The figures are well-crafted and greatly facilitate the understanding of the proposed framework's technical details.

S2. Addressing the targeted challenges, namely, LLM efficiency and the modeling of global semantic evolution in dynamic text, is essential to the DyTAG framework. Each component of the proposed design is carefully motivated and well justified.

S3. Extensive experimental results are provided, including the transductive & inductive performance, ablation and hyper-parameter study, as well as efficiency study, providing a comprehensive understanding of the effectiveness of the proposed framework.

**Weaknesses:**

W1. Several case studies are provided to better illustrate the reasoning process of the proposed framework, such as the summarization results generated by large language models.

W2. The authors employ a uniform partition strategy for global semantic summarization; however, since user intent may fluctuate over different time periods, this approach could be considered a technical limitation of the framework.

**Questions:**

None

---

> ### Author Response · Authors · 2025-11-17
> **Authors' Response**
>
> Thank you for your insightful comments. We provide detailed responses as follows.
>
> > *W1. Several case studies are provided to better illustrate the reasoning process of the proposed framework, such as the summarization results generated by large language models.*
>
> **A1**. Thank you for your valuable suggestion. We provide a case study from the Googlemap_CT dataset below, which demonstrates how the model uses global summarization to predict whether customer **Joe Golin** will visit the business **Liquor Depot** at timestamp 48264.
>
> > > ##### **Global Summarization for Customer Node "Joe Golin" (at t=47960)**
> > >
> > > Description: Joe Golin is a customer ***who values exceptional service, high-quality food, and unique dining experiences***, particularly in Asian cuisine, seafood restaurants, and ***pubs***. His enthusiasm for crab ragoons remains unwavering, with his positive reviews at Toshi Izakaya, J Timothy's Taverne, and 707 Asian Fusion showcasing his love for this dish. He is also a fan of imaginative waffles and great pizza, as evident from his positive reviews of THE SLOPPY WAFFLE LLC and Pizzeria Marzano. While he is willing to try new restaurants, his reviews suggest that he has high expectations and can be disappointed if they do not meet his standards, as seen in his review of Taco Bell. Joe also values excellent service, as evident from his positive reviews of Black Diamond Body Piercing, Mulligan's, O'Connor's Public House, and Onion Mountain Kitchen. His recent visits to Dos Amigos and Midas, where he experienced excellent service, further emphasize his appreciation for friendly and efficient service.
>
> > > ##### **Global Summarization for Business Node "Liquor Depot" (at t=42596)**
> > >
> > > Description: Liquor Depot, Inc. is a highly-recommended liquor store offering an extensive selection of beers, wines, and spirits at competitive prices. ***Their knowledgeable staff provides excellent assistance in finding the perfect products, frequently featuring a wide range of bargain wines and an exceptional bourbon selection, including rare and hard-to-find options.** The store continues to host events, such as the "Bourbon Bash," allowing customers to purchase limited-edition bottles. **They boast an impressive selection of 12 packs, wine, and whiskey, catering to various tastes and preferences***. The store is praised for its great prices, ***friendly staff***, and beautiful store environment, making it a go-to destination for liquor enthusiasts in Connecticut. The store's ability to adapt to changing tastes and trends, such as great St. Patrick's displays, further solidifies its reputation as a top choice.
>
> By processing these global summaries, our model can identify strong semantic alignments between the user's preferences and the business's offerings:
>
> 1. **Interest Alignment:** Joe Golin's summarized interest in "pubs" directly corresponds with "Liquor Depot" being a liquor store.
> 2. **Service Quality Alignment:** Joe's high value for "exceptional service" is matched by Liquor Depot's reputation for "knowledgeable" and "friendly staff" who provide "excellent assistance."
> 3. **Unique Experience Alignment:** Joe’s preference for "high-quality food and unique dining experiences" aligns with Liquor Depot’s "exceptional bourbon selection," "rare and hard-to-find options," and "impressive selection," which constitute a high-quality and unique product experience.
>
> Based on this semantic reasoning, the model can infer a high probability that Liquor Depot meets Joe Golin's criteria, thus predicting his visit at the future timestamp. This illustrates how global summarization provides crucial, high-level context that goes beyond individual interactions to improve predictive accuracy.
>
> We will expand this section in the revised manuscript with additional case studies and analysis to further strengthen the paper.
>
> > *W2. The authors employ a uniform partition strategy for global semantic summarization; however, since user intent may fluctuate over different time periods, this approach could be considered a technical limitation of the framework.*
>
> **A2**. Thank you for the insightful comment. We agree that user intent fluctuates and that relying solely on coarse-grained global summarization could overlook these dynamics. To tackle this potential issue, our DyGRASP framework integrates both **global** and **recent** temporal semantic reasoning. While the global summarization provides a long-term contextual view, our recent semantic reasoning module is specifically designed to capture the latest, short-term shifts in user intent. This component effectively addresses the fluctuations that the coarse-grained global summary might miss.
>
> Therefore, this multi-granularity approach to temporal semantics is a foundational strength of DyGRASP, allowing it to robustly model the very fluctuations you've highlighted. We will provide more clarifications in the revision.

---

> > ### Author Response · Authors · 2025-11-25
> > **Follow-up**
> >
> > Dear Reviewer,
> >
> >
> > Thank you for your insightful comments and positive evaluation. Please feel free to share any additional feedback you may have.
> >
> >
> > Best regards,
> > Authors

---

### Official Review · Reviewer_tZWv · 2025-10-31

**Soundness:** 2
**Presentation:** 2
**Contribution:** 2
**Rating:** 2
**Confidence:** 5

**Summary:**

DyGRASP tackles Dynamic Text-Attributed Graphs by combining large language model reasoning with temporal GNNs. It captures recent semantics via node-centric implicit reasoning and sliding windows, and global dynamics via explicit, prompt-guided summaries chained over time. Merging these signals with graph structure yields O(|E|) efficiency and accuracy, boosting Hit@10 by up to 34% and generalizing across LLMs and temporal GNN backbones.

**Strengths:**

1. The research topic is interesting and practically relevant, particularly the node retrieval task.

2. Efficiency is a critical issue, especially when employing LLMs as encoders.

**Weaknesses:**

1. There are multiple prior works on dynamic text-attributed graphs; however, the paper’s motivation is framed against static TAGs, which mispositions the contribution.

2. The related work is outdated—particularly for Temporal GNNs. The most recent citations end in 2023; please include the latest works.

3. The description of the “global reasoning chain” is unclear; stating it is “RNN-like” is insufficient and difficult to follow.

4. Although several dynamic TAG methods are cited, the paper does not compare against them in the evaluation.

**Major Concern:**

**(1) Empirically, DyGRASP (DyGFormer) with global and recent modules improves performance by only ~2% over DyGFormer alone in Tables 1–2, yet removing “global and recent” in Fig. 4 causes a >10% drop.**

**Questions:**

1. What is the training pipeline for the node-retrieval task—end-to-end or two-stage?

2. What is the rationale for “sorting by timestamp” in node-centric implicit reasoning?

3. In Eq. (2), the formulation seems to capture local rather than global semantics; please clarify how global semantics are obtained.

4. What do “gathering” and “aggregating” features mean in $MP_{G}(\cdot)$, which typically refers to the TGNNs? What is the difference between $MP_{G}(\cdot)$ and TGNNs' neighbor aggregation?

5. Could you please compare the proposed method with existing dynamic text-attributed graph approaches?

6. Could you please evaluate the effectiveness of the sliding window?

**Major Question:**

**(1) Empirically, DyGRASP (DyGFormer) with both global and recent modules improves performance by only ~2% over DyGFormer alone in Tables 1–2, yet removing “global and recent modules” in Fig. 4 causes a >10% drop. Could you please explain this discrepancy?**

---

> ### Author Response · Authors · 2025-11-17
> **Authors' Response [1/3]**
>
> Thank you for your valuable comments. We provide detailed responses as follows.
>
> > ***Major Concern & Question. Empirically, DyGRASP (DyGFormer) with both global and recent modules improves performance by only ~2% over DyGFormer alone in Tables 1–2, yet removing “global and recent modules” in Fig. 4 causes a >10% drop. Could you please explain this discrepancy?***
>
> **A1**. Thank you for this detailed observation. We have re-examined our results and would like to clarify that **there is no discrepancy. The results being compared correspond to two different tasks with distinct evaluation metrics.**
>
> Specifically, the **>10% performance drop in Figure 4** is from the ablation study on the **destination node retrieval** task, evaluated by Hit@10. In contrast, the **~2% improvement in Table 2** is for the **future link prediction** task, evaluated by AP/AUC. To demonstrate consistency, we can refer to the results for the same task. For instance, in the destination node retrieval task shown in Table 1 (Enron dataset, inductive setting), DyGRASP achieves a 12.66% absolute improvement over the DyGFormer baseline (95.18% vs 82.52%). This result from Table 1 is fully consistent with the >10% impact of our modules shown in Figure 4.
>
> To further clarify the paper, we will explicitly state the task ("destination node retrieval") in the caption of Figure 4 in our revised manuscript.
>
> > *W1. There are multiple prior works on dynamic text-attributed graphs; however, the paper’s motivation is framed against static TAGs, which mispositions the contribution.*
>
> **A2**. Thank you for your valuable feedback. We respectfully clarify that **our main contribution**—capturing overlooked recent-global spatio-temporal patterns via implicit and explicit LLM reasoning—**is specifically for dynamic ones.** As we discussed in RELATED WORK (lines 162-170), prior dynamic methods do not address these multi-granularity semantics or the associated reasoning efficiency challenges. We will revise our manuscript to frame our contributions to clarify our unique position.
>
> > *W2. The related work is outdated—particularly for Temporal GNNs. The most recent citations end in 2023; please include the latest works.*
>
> **A3**. Thank you for your constructive feedback. We will update our RELATED WORK section to include more recent TGNNs [1-12] such as SimpleDyG [1], PRES [2], mspace [3], CTAN [4] in the revision.
>
> > *W3. The description of the “global reasoning chain” is unclear; stating it is “RNN-like” is insufficient and difficult to follow.*
>
> **A4**. Thank you for your comments. We would like to clarify that the mechanism of the "global reasoning chain" is detailed in Section 4.2 and visually illustrated in Figure 2(b). Here we provide a further explanation for it.
>
> Firstly, the “global reasoning chain” is a sequential update process. For each time segment, we instruct the LLM to generate a new, updated textual description for a node by providing it with two key pieces of information: 1) the node's *previous* summary description, and 2) the set of *new* interactions that occurred within that segment.
>
> Secondly, we describe this as "RNN-like" because the output of one step (the newly generated summary) serves as the input for the subsequent step, creating a chain that propagates long-term semantic information. This design allows the model to efficiently capture the evolution of a node's global semantics over time, mirroring an RNN's ability to maintain a state and process sequences without needing to re-process the entire history at each step. This ensures both effectiveness and low computational overhead.
>
> Following your suggestions, we will further revise the descriptions.
>
> > *W4&Q5. Although several dynamic TAG methods are cited, the paper does not compare against them in the evaluation. Could you please compare the proposed method with existing dynamic text-attributed graph approaches?*
>
> **A5**. Thank you for the suggestion. First, we would like to clarify that **these methods are concurrent preprints by the submission deadline** and according to the ICLR guideline **authors are not required to compare to papers solely on arXiv**. Secondly, we would clarify that making a direct comparison with them is challenging due to the following issues:
>
> - **CROSS** uses a different dataset train:val:test split ratio and metric than the DTGB benchmark.
> - **LKD4DyTAG** employs a base GNN not available in the benchmark and uses a different batch size and learning rate.
> - **GAD** evaluates on only a subset of the data due to LLM efficiency constraints.
>
> Due to the lack of open-source code for these works, it is difficult for us to evaluate them under the same settings. On the other hand, our strict adherence to the DTGB setup ensures reproducibility and a fair comparison against established baselines. We would extending the comparison when the source codes of these methods are available.

---

> ### Author Response · Authors · 2025-11-17
> **Authors' Response [2/3]**
>
> > *Q1. What is the training pipeline for the node-retrieval task—end-to-end or two-stage?*
>
> **A6**. Thanks for the question. The training for the node-retrieval task is **end-to-end**, which is the same as future link prediction task. As detailed in the "Training for DyGRASP" paragraph within Section E, following DTGB, our model directly predicts the probability of an interaction between any two given nodes. We then supervise the model's training using a binary cross-entropy loss.
>
> The distinction between these two tasks lies in the **inference phase**. Unlike future link prediction, which simply predicts whether an interaction will occur, the node-retrieval task requires the model to respectively provide the interaction probabilities from the source node to a series of destination nodes, which are then manually ranked and evaluated by the following procedure. This evaluation protocol is also thoroughly explained in Section E.
>
> > *Q2. What is the rationale for “sorting by timestamp” in node-centric implicit reasoning?*
>
> **A7**. Thanks for the question. The primary rationale for "sorting by timestamp" is to **align the processing of interaction sequences with the inherent unidirectional attention mechanism of LLMs to prevent future information leakage.**
>
> Specifically, LLMs process text sequentially using a causal attention mechanism, where a token at any given position can only attend to the tokens that precede it. This mechanism's unidirectionality is perfectly analogous to the forward-only progression of time. As detailed on lines 213-215 and illustrated in Figure 2(a), when we arrange a node's historical interactions by sorting them based on their timestamps, we ensure that an interaction occurring later in time is also placed at a later position in the input sequence. Consequently, the LLM is constrained to capture the temporal semantics for any given interaction based *only* on past interactions, effectively preventing it from accessing information from the future.
>
> This alignment is key to our method's efficiency, as it enables the extraction of recent temporal semantic features for all interactions within the sequence in a **single forward pass**, without corrupting the temporal dependencies. We will revise the manuscript to clarify this point.
>
> > *Q3. In Eq. (2), the formulation seems to capture local rather than global semantics; please clarify how global semantics are obtained.*
>
> **A8**. Thanks for the question. We would like to clarify how global semantics are captured in two respects.
>
> First, a single computation of Eq. (2) is inherently more global than local. As detailed in Section 4.2, the input $S_i$ represents the set of all interactions within the time interval from $\\hat{t}_{i-1}$ to $\\hat{t}_i$. This temporal span, $\\delta^g = \\hat{t}\_i - \\hat{t}\_{i-1}$, is significantly larger than the span $\\delta^r$ which characterizes short-term, local semantics ($\\delta^g >> \\delta^r$). Therefore, Eq. (2) processes a temporally global span of interactions in each step.
>
> Second, global semantics are captured and propagated through multiple computations. The formulation in Eq. (2) can be unrolled as a recurrent, RNN-like inference chain:
>
> $D_i = \\text{LLM}(D_{i-1}, S_i) = \\text{LLM}(\\text{LLM}(D_{i-2}, S_{i-1}), S_i) = \\text{LLM}(\\text{LLM}(\\text{LLM}(D_{i-3}, S_{i-2}), S_{i-1}), S_i) =\\ ...$
>
> This structure ensures that the output $D_i$ encapsulates not only the semantics of the current global interaction set $S_i$ but also recursively incorporates the semantics from all preceding global interaction sets $\\{S_1, S_2, ..., S_{i-1}\\}$. Consequently, this mechanism allows the model to build a comprehensive understanding of global semantics over the entire history of interactions. We will revise the manuscript to clarify this point.
>
> > *Q4. What do “gathering” and “aggregating” features mean in $\\text{MP}\_{\\mathcal{G}}(\\cdot)$, which typically refers to the TGNNs? What is the difference between $\\text{MP}\_{\\mathcal{G}}(\\cdot)$ and TGNNs' neighbor aggregation?*
>
> **A9**. Thanks for the question. **$\\text{MP}\_{\\mathcal{G}}(\\cdot)$ precisely represents the neighbor aggregation process of the TGNNs**, as stated in lines 319-323.  There is no difference between $\\text{MP}\_{\\mathcal{G}}(\\cdot)$ and TGNNs' neighbor aggregation; it is just an abstract and unified notation for this process.
>
> We introduced this notation to emphasize the modularity of our Graph Structure Layer. $\\text{MP}\_{\\mathcal{G}}(\\cdot)$ serves as a general placeholder for the process of "gathering and aggregating features from neighbors", which can be instantiated by any TGNNs (e.g., DyGFormer, TGAT). In this layer, we build upon the integrated feature $\\text{M}$ by applying the TGNN's specific structural aggregation mechanism (denoted by $\text{MP}\_{\\mathcal{G}}(\\cdot)$) to enable a deeper fusion between semantic and structural features. We will revise the manuscript to clarify this point.

---

> ### Author Response · Authors · 2025-11-17
> **Authors' Response [3/3]**
>
> > *Q6. Could you please evaluate the effectiveness of the sliding window?*
>
> **A10**. Thanks for the question. The sliding window mechanism is primarily designed for **efficiency and practical feasibility**. A direct ablation by removing the window is computationally infeasible. Thus, we clarify the effectiveness through the following analysis:
>
> 1.  **Computational Challenges**: Removing the sliding window would require inputting a node's entire interaction history into the LLM at once. This approach is impractical due to:
>     *   **Exceeding LLM Context Length**: The full history of interactions for many nodes would generate input text far exceeding the context window limits of current LLMs.
>     *   **Prohibitive Resource Consumption**: Processing such long sequences would demand excessive computational and memory resources. As the number of interactions grows dynamically, the ever-increasing length of historical data would inevitably lead to out-of-memory errors, making inference impossible.
>
> 2.  **Performance Degradation**: We can also infer that removing the sliding window would likely harm performance. As widely observed, and also cited in our paper (lines 456-457), LLM performance tends to degrade as input length increases significantly. Our results in Figure 5 provide direct evidence supporting this: the performance of recent temporal semantic reasoning declines after the window size $c$ exceeds 16. Removing the window is equivalent to setting $c$ to an unbounded value, which would predictably lead to a substantial performance drop.
>
> In summary, the sliding window is a necessary component that ensures our model is both scalable and effective. Our analysis strongly suggests that its removal would be detrimental to both computational feasibility and model performance. We will add the discussion in the revision.
>
> [1] Wu, Yuxia, Yuan Fang, and Lizi Liao. "On the feasibility of simple transformer for dynamic graph modeling." Proceedings of the ACM Web Conference 2024. 2024.
>
> [2] Su, Junwei, Difan Zou, and Chuan Wu. "PRES: Toward Scalable Memory-Based Dynamic Graph Neural Networks." *The Twelfth International Conference on Learning Representations*, 2024.
>
> [3] Rahman, Aniq Ur, and Justin Coon. "Node Feature Forecasting in Temporal Graphs: an Interpretable Online Algorithm." Transactions on Machine Learning Research. 2025.
>
> [4] Gravina, Alessio, et al. "Long range propagation on continuous-time dynamic graphs." *Proceedings of the 41st International Conference on Machine Learning*. 2024.
>
> [5] Zhang, Siwei, et al. "Towards adaptive neighborhood for advancing temporal interaction graph modeling." Proceedings of the 30th ACM SIGKDD Conference on Knowledge Discovery and Data Mining. 2024.
>
> [6] Chen, Xi, et al. "Dtformer: A transformer-based method for discrete-time dynamic graph representation learning." Proceedings of the 33rd ACM International Conference on Information and Knowledge Management. 2024.
>
> [7]Xu, Yuanyuan, et al. "UniDyG: A Unified and Effective Representation Learning Approach for Large Dynamic Graphs." IEEE Transactions on Knowledge and Data Engineering (2025).
>
> [8]Liu, Hanwen, et al. "SALoM: Structure Aware Temporal Graph Networks with Long-Short Memory Updater." The Thirty-ninth Annual Conference on Neural Information Processing Systems. 2025.
>
> [9]Li, Yingxuan, et al. "Ranking on dynamic graphs: An effective and robust band-pass disentangled approach." Proceedings of the ACM on Web Conference 2025. 2025.
>
> [10]Xu, Yuanyuan, et al. "TimeSGN: Scalable and effective temporal graph neural network." 2024 IEEE 40th International Conference on Data Engineering (ICDE). IEEE, 2024.
>
> [11]Liu, Hanmo, et al. "A Selective Learning Method for Temporal Graph Continual Learning." Forty-second International Conference on Machine Learning. 2025.
>
> [12]Li, Hao, et al. "TMetaNet: Topological Meta-Learning Framework for Dynamic Link Prediction." Forty-second International Conference on Machine Learning. 2025.

---

> ### Comment · Reviewer_tZWv · 2025-11-18
> **Comments after rebuttal**
>
> For **A1**, the authors initially did not illustrate the TGNNs (DyGFormer) used in the ablation study. After the rebuttal, this major concern has been resolved. However, the authors should carefully describe all experimental settings for clarity.
>
> For **A3, A4, A6, A7, A10**, the authors’ response addresses my questions, provided these issues are properly fixed in the revised version.
>
> For **A2**,  **this concern still remains as this paper focuses on dynamic ones and there are several dynamic works.**
>
> For **A5**, LKD4DyTAG (AAAI 2025, Jan 2025) was published well before the submission deadline. **Thus, this concern still remains.**
>
> For **A8**, I double-checked Equation 2 and Section 4.2 and did not find any explanation there. I still cannot understand how the global semantics are derived from Equation 2.
>
> For **A9**, if "There is no difference between $Mp_G()$and TGNNs' neighbor aggregation", then Equation 7 is not entirely correct.

---

> > ### Author Response · Authors · 2025-11-19
> > **Authors' Follow-up Response [1/2]**
> >
> > Thank you for your prompt reply! We are pleased to know our rebuttal has addressed your major concern and more than half of your questions. For your remaining comments, we provide detailed responses as follows.
> >
> > > A2&A5.
> >
> > Thank you for your valuable comments. Following your suggestion, we provide a comparison with LKD4DyTAG (AAAI 2025 workshop). Note that the source code of LKD4DyTAG is not available, so we directly compare with their reported results. The results are presented below.
> >
> > |      |                     |           | Transductive |              |           |           | Inductive |              |           |
> > | ---- | ------------------- | --------- | ------------ | ------------ | --------- | --------- | --------- | ------------ | --------- |
> > |      |                     | GDELT     | Enron        | Googlemap_CT | Avg.      | GDELT     | Enron     | Googlemap_CT | Avg.      |
> > |      | LKD4DyTAG           | 97.76     | **99.43**    | 91.44        | 96.21     | **93.69** | **95.50** | 88.90        | 92.70     |
> > | AP   | DyGRASP (TGAT)      | **97.78** | 98.04        | **96.23**    | **97.35** | 93.05     | 91.66     | **95.31**    | **93.34** |
> > |      | DyGRASP (DyGFormer) | 97.40     | 98.81        | 94.85        | 97.02     | 92.23     | 95.03     | 93.68        | 93.65     |
> > |      |                     |           |              |              |           |           |           |              |           |
> > |      | LKD4DyTAG           | **99.98** | **98.87**    | 91.27        | 96.71     | **95.77** | 93.67     | 88.57        | 92.67     |
> > | AUC  | DyGRASP (TGAT)      | 97.22     | 98.34        | 94.87        | 96.81     | 90.74     | 92.38     | 93.47        | 92.20     |
> > |      | DyGRASP (DyGFormer) | 97.60     | 98.86        | **94.89**    | **97.12** | 92.19     | **94.85** | **93.58**    | **93.54** |
> >
> > As shown in the table, on the average results across three datasets, DyGRASP consistently outperforms LKD4DyTAG for both AP and AUC under both transductive and inductive settings. A more detailed analysis reveals the specific strengths of our method:
> >
> > - On datasets with less textual information like GDELT, where graph structure is more critical, LKD4DyTAG performs slightly better. This is likely due to its custom-designed base TGNN being more effective at capturing structural features.
> > - However, on text-rich datasets with complex semantic patterns like Googlemap_CT, DyGRASP demonstrates a significant lead. This validates our core contribution: DyGRASP's explicit design for modeling recent and global temporal semantics provides superior semantic reasoning capabilities compared to existing methods.
> >
> > Based on these additional experiments, we would like to further clarify our contributions compared to existing dynamic methods. While prior dynamic works exist, they do not address the central problem we tackle: modeling the distinct, multi-granularity temporal patterns in DyTAGs. Our novelty is the introduction of a framework that leverages both implicit LLM reasoning for *recent* temporal dependencies and explicit reasoning for *global* semantic evolution. This specific "recent-global" formulation and our efficient solution are what distinguish our work. This new comparison also verifies the effectiveness of our approach, particularly in the rich semantic environments that DyTAGs often represent. We will revise the paper to highlight this contrast with other dynamic approaches more clearly.

---

> > ### Author Response · Authors · 2025-11-19
> > **Authors' Follow-up Response [2/2]**
> >
> > > A8.
> >
> > Thank you for your valuable comments. We would like to clarify that the key lies in the **cumulative, RNN-like reasoning chain** of Eq. (2).
> >
> > Specifically, let's first define the symbols in Eq. (2): $D_i = \text{LLM}(D_{i-1}, S_i)$.
> > *   As stated in lines 262-266, **$S_i$** is the chronologically ordered sequence of all interactions occurring within a specific time segment *$t_{i-1}\sim t_{i}$*.
> > *   As described in lines 269-271, **$D_i$** is the LLM-generated textual description that summarizes the node’s dynamic features corresponding to the period of $S_i$.
> >
> > The process for acquiring global semantics is designed to be **cumulative and iterative**:
> >
> > 1.  **Cumulative Summary**: Crucially, the input $D_{i-1}$ is not a local feature but a textual summary of the node's **entire interaction history** up to the start of the current segment $S_{i-1}$.
> > 2.  **Iterative Update**: The LLM is prompted to **update** this historical summary ($D_{i-1}$) by integrating the new interactions from the current segment ($S_i$). For instance, $D_1$ summarizes $S_1$; $D_2$ then integrates the information from $S_2$ into $D_1$ to summarize the combined history of $S_1$ and $S_2$, and so on.
> >
> > Consequently, the final description $D_s$ encapsulates the semantics from the node's **entire history** (from $S_1$ through $S_s$), effectively capturing its long-term, global temporal evolution. "Global" in this context refers to the full temporal scope of a node's activities.
> >
> > We will incorporate the above discussions in a revised version.
> >
> > > A9.
> >
> > Thank you for your insightful comment. We appreciate the opportunity to clarify the formulation of Equation (7). In general, the computation of a TGNN can be abstracted into two main stages:
> >
> > 1. **Neighbor Aggregation**: For a given node $v$ and time $t$, the model aggregates features from $v$'s neighbors based on interactions occurring before $t$. This process, which captures the dynamic graph structure, is represented by **$\text{MP}_{\mathcal{G}}(v,t)$** in our equation.
> > 2. **Feature Merging**: The aggregated structural feature is then merged with the node $v$'s own representation, $\text{M}$, to produce an updated node embedding. This is represented by the broader **$\text{TGNN}(\text{M}, \text{MP}_{\mathcal{G}}(v,t))$** function in our original equation.
> >
> > The primary purpose of Equation (7) is to illustrate how DyGRASP enables a learnable, deep fusion of our semantically-enriched node features ($\text{M}$) with the dynamic structural features ($\text{MP}_{\mathcal{G}}(v,t)$) derived from the graph.
> >
> > To further clarify this distinction, we will revise Equation (7) in the manuscript to $\text{Merge}^{\small{\text{TGNN}}}(\text{M}, \text{MP}_{\mathcal{G}}(v,t))$ and explicitly add a brief explanation of this merge process. We believe this modification will more accurately reflect the specific roles of each component.
> >
> > Thank you again for your insightful comments, which help improve our paper. Please let us know if you have further comments.

---

> > > ### Author Response · Authors · 2025-11-25
> > > **Follow-up**
> > >
> > > Dear Reviewer,
> > >
> > > Thank you again for your insightful comments. We have provided additional responses and results for your remaining points (A2, A5, A8, A9) and hope they fully address your concerns.
> > >
> > > As our previous rebuttal has already resolved your major concern (A1) and most of the comments (A3, A4, A6, A7, A10), we believe the manuscript is now significantly strengthened. We hope you find the revised manuscript worthy of an improved score.
> > >
> > > Please let us know if you have any further feedback.
> > >
> > > Best regards,
> > > Authors

---

> > > > ### Author Response · Authors · 2025-12-01
> > > > **Discussion Summary**
> > > >
> > > > We sincerely appreciate the hard work of all ACs, SACs and PCs. For your convenience, we would like to briefly summarize our interactions with the reviewer. The reviewer initially raised 10 questions, including one "major concern". Most of these issues stemmed from misunderstandings of our paper or some of our arguments were not precise enough. After our rebuttal, the reviewer **publicly acknowledged that 6 out of the 10 questions had been addressed, including the major concern**. For the remaining four questions, we provided further clarifications, **including the additional experiments the reviewer requested**. Unfortunately, the reviewer did not have the opportunity to respond. We have provided a revision that incorporates all these changes.
> > > >
> > > >  We hope this summary helps reduce the burden if another AC takes over our paper. Please let us know if you have any further questions or comments.
> > > >
> > > > Best regards,
> > > > Authors

---

### Official Review · Reviewer_hKYq · 2025-10-31

**Soundness:** 3
**Presentation:** 3
**Contribution:** 3
**Rating:** 6
**Confidence:** 4

**Summary:**

This paper introduces DyGRASS, a framework designed to address semantic comprehension and reasoning in dynamic text-attributed graphs. The authors design a sliding window strategy that effectively captures short-term temporal patterns. Additionally, they employ an LLM to distill long-range, chain-structured semantic information at the node level, enabling comprehensive global semantic representation learning. Evaluations conducted on four benchmark datasets demonstrate the model's outstanding performance and its adaptability to different LLM and temporal GNN backbones.

**Strengths:**

1. The inefficiency of LLMs and the limited semantic modeling capability of GNNs represent two major challenges in modeling DyTAGs. This paper offers an effective and well-founded solution to address these issues.

2. The technical design of the proposed framework is well-motivated, and its effectiveness has been empirically validated through experiments.

3. Comprehensive experimental results are provided, demonstrating the superior performance and efficiency of the proposed framework.

**Weaknesses:**

1. To better validate the efficiency of the proposed DyGRASS, a comparison of inference time and token consumption with other baseline models should be provided.

2. Case studies illustrating the global summarization results of LLMs are recommended, along with an analysis of how global semantic information enhances prediction performance.

**Questions:**

See above.

---

> ### Author Response · Authors · 2025-11-17
> **Authors' Response [1/2]**
>
> Thank you for your insightful comments. We provide detailed responses as follows.
>
> >  *W1. To better validate the efficiency of the proposed DyGRASS, a comparison of inference time and token consumption with other baseline models should be provided.*
>
> **A1**. Thank you for your valuable suggestion regarding the efficiency of DyGRASP.  As many of our baselines are TGNNs, token consumption is not a universally applicable metric for a fair comparison. Therefore, we present the inference speed (edges/second) below to evaluate all methods on a common basis.
>
> | Inference Speed (edges/s) | Enron   | Googlemap_CT |
> | ------------------------- | ------- | ------------ |
> | JODIE                     | 3,372   | OOM          |
> | DyRep                     | 2,872   | OOM          |
> | CAWN                      | 313     | 307          |
> | TCL                       | 1,157   | 2,621        |
> | GraphMixer                | 2,410   | 3,098        |
> | TGAT                      | 587     | 545          |
> | DyGFormer                 | 1,077   | 1,165        |
> | Llama-3.1                 | 15      | 13           |
> | **DyGRASP (Ours)**        | **729** | **757**      |
>
> Despite incorporating an LLM, DyGRASP's inference speed is competitive with other TGNNs and is notably faster than complex attention-based (e.g., TGAT) and random-walk-based (e.g., CAWN) models. This speed is orders of magnitude faster than that of a pure LLM baseline (e.g., Llama 3.1), demonstrating the high efficiency of our method at inference time. As we discussed in the "Efficiency Analysis" section (Lines 458-462), DyGRASP decouples the LLM reasoning from the online prediction task, making the inference phase as fast as a lightweight TGNN. The results in the table also confirm this. We will add this comparison to our revised paper.

---

> ### Author Response · Authors · 2025-11-17
> **Authors' Response [2/2]**
>
> > *W2. Case studies illustrating the global summarization results of LLMs are recommended, along with an analysis of how global semantic information enhances prediction performance.*
>
> **A2**. Thank you for your valuable suggestion. Following your suggestion, we provide a case study from the Googlemap_CT dataset below, which demonstrates how the model uses global summarization to predict whether customer **Joe Golin** will visit the business **Liquor Depot** at timestamp 48264.
>
> > > ##### **Global Summarization for Customer Node "Joe Golin" (at t=47960)**
> > >
> > > Description: Joe Golin is a customer ***who values exceptional service, high-quality food, and unique dining experiences***, particularly in Asian cuisine, seafood restaurants, and ***pubs***. His enthusiasm for crab ragoons remains unwavering, with his positive reviews at Toshi Izakaya, J Timothy's Taverne, and 707 Asian Fusion showcasing his love for this dish. He is also a fan of imaginative waffles and great pizza, as evident from his positive reviews of THE SLOPPY WAFFLE LLC and Pizzeria Marzano. While he is willing to try new restaurants, his reviews suggest that he has high expectations and can be disappointed if they do not meet his standards, as seen in his review of Taco Bell. Joe also values excellent service, as evident from his positive reviews of Black Diamond Body Piercing, Mulligan's, O'Connor's Public House, and Onion Mountain Kitchen. His recent visits to Dos Amigos and Midas, where he experienced excellent service, further emphasize his appreciation for friendly and efficient service.
>
> > > ##### **Global Summarization for Business Node "Liquor Depot" (at t=42596)**
> > >
> > > Description: Liquor Depot, Inc. is a highly-recommended liquor store offering an extensive selection of beers, wines, and spirits at competitive prices. ***Their knowledgeable staff provides excellent assistance in finding the perfect products, frequently featuring a wide range of bargain wines and an exceptional bourbon selection, including rare and hard-to-find options.** The store continues to host events, such as the "Bourbon Bash," allowing customers to purchase limited-edition bottles. **They boast an impressive selection of 12 packs, wine, and whiskey, catering to various tastes and preferences***. The store is praised for its great prices, ***friendly staff***, and beautiful store environment, making it a go-to destination for liquor enthusiasts in Connecticut. The store's ability to adapt to changing tastes and trends, such as great St. Patrick's displays, further solidifies its reputation as a top choice.
>
> By processing these global summaries, our model can identify strong semantic alignments between the user's preferences and the business's offerings:
>
> 1. **Interest Alignment:** Joe Golin's summarized interest in "pubs" directly corresponds with "Liquor Depot" being a liquor store.
> 2. **Service Quality Alignment:** Joe's high value for "exceptional service" is matched by Liquor Depot's reputation for "knowledgeable" and "friendly staff" who provide "excellent assistance."
> 3. **Unique Experience Alignment:** Joe’s preference for "high-quality food and unique dining experiences" aligns with Liquor Depot’s "exceptional bourbon selection," "rare and hard-to-find options," and "impressive selection," which constitute a high-quality and unique product experience.
>
> Based on this semantic reasoning, the model can infer a high probability that Liquor Depot meets Joe Golin's criteria, thus predicting his visit at the future timestamp. This illustrates how global summarization provides crucial, high-level context that goes beyond individual interactions to improve predictive accuracy.
>
> We will expand this section in the revised manuscript with additional case studies and analysis to further strengthen the paper.

---

> > ### Comment · Reviewer_hKYq · 2025-11-24
> >
> > Thanks for the authors' feedback. I would like to maintain my current positive score.

---

> > > ### Author Response · Authors · 2025-11-25
> > > **Response**
> > >
> > > Dear Reviewer,
> > >
> > >
> > > Thank you for your insightful comments and positive evaluation. Please feel free to share any additional feedback you may have.
> > >
> > >
> > > Best regards,
> > > Authors

---

### Author Response · Authors · 2025-12-01

We sincerely thank the reviewers for their dedicated time and effort in evaluating our manuscript. Their constructive criticisms and insightful comments have been instrumental in enhancing the quality of this work. In response to the reviewers' suggestions, we would like to highlight the key changes made in the revised manuscript,

**Key Changes Made**

1. We have cited more recent TGNNs and further clarified our unique position against existing LLM for DyTAGs studies in RELATED WORK.
2. We have revise Eq. (7) to $\text{Merge}^{\small{\text{TGNN}}}(\text{M}, \text{MP}_{\mathcal{G}}(v,t))$ and explicitly add a brief explanation of this merge process.
3. We have clarified the main experimental tasks of this paper and the main base TGNNs adopted by DyGRASP in Section 5.1.
4. We have added a comparison of *online prediction efficiency*  with baselines in Section 5.3, where we show the inference phase  of DyGRASP as fast as a lightweight TGNN.
5. We have provided a comparison against existing LLM-for-DyTAGs method in Appendix E.4.
6. We have added Appendix F with the CASE STUDY to demonstrates how DyGRASP uses global summarization to predict future links.

We have updated our paper with the revisions in the revised PDF. We believe these revisions address the concerns raised by the reviewers and enhance the quality of the paper. We sincerely appreciate the hard work of all ACs, SACs, and PCs for their timely response after the incident and significant contributions to ICLR community.

Best regards,

Authors

---

### Meta-Review · Area_Chair_4zAD · 2026-01-13

**Summary:**

This paper receives three positive scores and one negative score. Reviewer hKYq concerns its inference and token consumption efficiency. Reviewer tZWv concerns its outdated related work and unclear term description. The comparing algorithms are also not sufficient. Reviewer amvQ concerns its uniform partition strategy for global semantic summarization. Reviewer jwom concerns its computational costs.

Despite these defects, Reviewer hKYq acknowledges its well-motivated technical design. Reviewer tZWv acknowledges that the research topic is interesting. Reviewer amvQ acknowledges its extensive experiments. Reviewer jwom acknowledges its novelty of dual learning approach. After rebuttal, Reviewers hKYq and jwom express that they would like to maintain the original positive scores. Most concerns of Reviewer tZWv have been addressed.   Reviewer tZWv still concerns that the latest algorithm, LKD4DyTAG, is not compared, and Equation 7 is not entirely correct. How the global semantics are derived from Equation 2 is also unclear.

**Reviewer Concerns:**

After rebuttal, most concerns have been addressed. Reviewer tZWv still concerns that the latest algorithm, LKD4DyTAG, is not compared, and Equation 7 is not entirely correct. Besides, how the global semantics are derived from Equation 2 is unclear.

**Reviewer Scores:**

Reviewers hKYq and jwom express that they would like to maintain the original positive scores. Reviewer tZWv may maintain the original score since he/she express that there are still some concerns unsolved. Reviewer amvQ may maintain the original score 8, since the authors provide  a case study  to predict whether customer Joe Golin will visit the business Liquor Depot.

---

### Decision · Program_Chairs · 2026-01-26

Accept (Poster)